# H2B ubiquitination recruits FACT to maintain a stable altered nucleosome state for transcriptional activation

Anfeng Luo[1,6], Jingwei Kong[2,3,6], Jun Chen[1,6], Xue Xiao[2,6], Jie Lan[4], Xiaorong Li[4], Cuifang Liu[4], Peng-Ye Wang[2,3,5], Guohong Li[3,4], Wei Li[2,4,5] ✉ & Ping Chen[1,4] ✉

Histone H2B mono-ubiquitination at lysine 120 (ubH2B) has been found to regulate transcriptional elongation by collaborating with the histone chaperone FACT (Facilitates Chromatin Transcription) and plays essential roles in chromatin-based transcriptional processes. However, the mechanism of how ubH2B directly collaborates with FACT at the nucleosome level still remains elusive. In this study, we demonstrate that ubH2B impairs the mechanical stability of the nucleosome and helps to recruit FACT by enhancing the binding of FACT on the nucleosome. FACT prefers to bind and deposit H2A-ubH2B dimers to form an intact nucleosome. Strikingly, the preferable binding of FACT on ubH2B-nucleosome greatly enhances nucleosome stability and maintains its integrity. The stable altered nucleosome state obtained by ubH2B and FACT provides a key platform for gene transcription, as revealed by genome-wide and time-course ChIP-qPCR analyses. Our findings provide mechanistic insights of how ubH2B directly collaborates with FACT to regulate nucleosome dynamics for gene transcription.

Eukaryotic DNA is packaged into chromatin by histones. Nucleosome, the basic structural unit of chromatin, is composed of a histone octamer (two copies of each H2A, H2B, H3 and H4) wrapped by 147 base pairs (bp) of DNA about 1.7 turns in a left-handed manner[1,2]. Generally, the nucleosome acts as a repressor to limit the access of DNA and provides a structural platform for DNA-related processes including DNA transcription and replication, in which the nucleosome barrier needs be temporarily displaced to expose DNA, and then rapidly restored afterwards to preserve the original epigenetic identity[3,4]. During the transcriptional elongation process, the nucleosome structure has been found to be dynamically regulated, with the nucleosome disassembled in front of transcribing RNA polymerase II (Pol II) and rapidly reassembled in its wake. Recent investigations have implicated

that post-translational modifications (PTMs) of core histones play a critical role in the process. One such typical modification is the mono-ubiquitination of H2B C terminus at lysine 120 in human (ubH2B, occurs at lysine 123 in budding yeast), in which the bulky 76 amino-acid protein ubiquitin of approximately 8.5 kDa is covalently attached through its C-terminus to the ε-amino group of lysine residue[5]. Histone H2B is transiently ubiquitinated and deubiquitinated at coding region of actively transcribed genes[5]. In mammalian, H2B mono-ubiquitination is catalyzed primarily by heteromeric RNF20/40 complex[6,7] and deubiquitinated by deubiquitinating enzyme (DUB) including several ubiquitin-specific proteases (USPs) as USP12, USP22, USP44 and USP46 etc[8–10]. Aberrant ubH2B levels have been shown to affect cell differentiation and development, and linked with a variety of cancers[11,12].

[1]Department of Immunology, School of Basic Medical Sciences, Beijing Key Laboratory for Tumor Invasion and Metastasis, Capital Medical University, Beijing 100069, China. [2]National Laboratory for Condensed Matter Physics and Key Laboratory of Soft Matter Physics, Institute of Physics, Chinese Academy of Sciences, Beijing 100190, China. [3]University of Chinese Academy of Sciences, Beijing 100049, China. [4]National Laboratory of Biomacromolecules, CAS Center for Excellence in Biomacromolecules, Institute of Biophysics, Chinese Academy of Sciences, Beijing 100101, China. [5]Songshan Lake Materials Laboratory, Dongguan, Guangdong 523808, China. [6]These authors contributed equally: Anfeng Luo, Jingwei Kong, Jun Chen, Xue Xiao. ✉e-mail: weili007@iphy.ac.cn; chenping@ccmu.edu.cn

H2B mono-ubiquitination has long been proposed to play an important role in gene transcription. ubH2B is broadly associated with transcribed region of highly expressed genes in human cells[13]. With a bulky moiety addition onto histones, ubH2B has been proposed to disrupt the folding of higher-order chromatin structure and lead to an open and biochemically accessible fiber conformation by in vitro analysis of nucleosome arrays containing semisynthetic ubH2B[14]. Although a mild nucleosome destabilizing effect of ubH2B was measured by in vitro Nap1-assited FRET (Fluorescence Resonance Energy Transfer)-based assay[15], Chandrasekharan et al. showed that the nucleosome stability is enhanced when the level of ubH2B increases by using a repertoire of biochemical and genetic analyses in yeast cells[16]. In addition, single-molecule DNA unzipping invesigations by optical tweezers showed that ubH2B strengthened the proximal dimer region of nucleosome and increased the overall barrier strength of Pol II transcription through a nucleosome[17]. ubH2B has also been found to be required for efficient reassembly of nucleosomes at gene bodies in the wake of Pol II-mediated transcription elongation[18]. Nevertheless, how ubH2B directly affects the nucleosome dynamics still remains to be answered.

The FACT complex, who earns the name "Facilitates Chromatin Transcription" from its ability to allow RNA Pol II to transcribe DNA templates at the level of chromatin, has been characterized to play an essential role in the chromatin-related processes[19]. The highly conserved FACT complex is initially identified as a chaperone of histone H2A/H2B dimer[19,20], and essential for cell viability[21,22]. FACT has been found to not only facilitate the progression of DNA and RNA polymerases on chromatin templates[19,20], but also maintain the genome-wide integrity of chromatin[23–25]. Recent in vitro structural and single-molecular investigations have revealed the dual functions of FACT on nucleosome by facilitating both the nucleosome disassembly and reassembly[26,27]. FACT has been found to tether both the (H3/H4)$_2$ tetramer and H2A/H2B dimer on DNA by its two subunits SPT16 and SSRP1, which function distinctly but coordinately to fulfill these apparently opposite functions of FACT on nucleosome[26,27]. It is of great interest to investigate how the FACT's functions are regulated by different epigenetic factors, as for histone mono-ubiquitination. The H2A mono-ubiquitination at lysine 119 (ubH2A) does not affect the FACT's chaperone function in nucleosome assembly, but greatly blocks the FACT's function in nucleosome disassembly[28]. Different from ubH2A, ubH2B presents unique interactions with FACT for gene transcription. Previous analysis by in vitro transcription assays has showed that ubH2B can cooperate with the FACT complex to stimulate the efficiency of elongation by RNA Pol II, probably by the efficient displacement of H2A/H2B dimer from nucleosome[29]. But to date, how ubH2B directly regulates the function of FACT on nucleosome dynamics are still poorly understood.

In this work, we focus on the regulation of ubH2B on nucleosome dynamics and its effect on the function of FACT complex. We reveal that the mono-ubiquitination on H2B impairs the stability of nucleosome. ubH2B facilitates the recruitment of FACT by enhancing the binding of FACT on nucleosome, and FACT greatly enhances the stability of ubH2B-nucleosome and helps to maintain nucleosome integrity. In addition, ubH2B promotes the binding of FACT on H2A/H2B dimer, maintains and enhances the FACT's chaperone function to deposit H2A/H2B dimer to form intact nucleosome. The genome-wide study and time-course ChIP-qPCR analyses show that ubH2B helps to recruit FACT to maintain nucleosome integrity and facilitate gene transcription. Our findings provide mechanistic insights of how H2B ubiquitination directly collaborates with FACT to regulate nucleosome dynamics during the chromatin-related transcription process.

## Results

### ubH2B impairs the mechanical stability of nucleosome

The effect of ubH2B on the stability and folding kinetics of nucleosome was directly investigated by single-molecule magnetic tweezers. We traced the dynamics of force-dependent conformational transition of mono-nucleosome reconstituted in vitro based on a 409 bp DNA containing one Widom 601 nucleosome positioning sequence in the middle (Fig. 1). The reconstituted mono-nucleosomes containing ubH2B or unmodified H2B were characterized by gel shift analysis and AFM imaging, which showed that the nucleosomes are well positioned in the middle of DNA (Fig. 1a, b). To investigate the mechanical stability of nucleosome, we traced its unfolding dynamics under a continuously increasing tension by tuning the magnets in $z$ direction at 10 μm/s (Fig. 1c). The stretching force was ranged from 0.1pN to 30pN for one trajectory and repeated for at least three times for each nucleosome sample to quantitatively assess the stability and reversibility of unmodified H2B and ubH2B nucleosomes (Fig. 1d, e). For the nucleosome with unmodified H2B (H2B-nucleosome), the typical two-step unfolding process was revealed at rupture forces around 4pN and 23pN, corresponding to the unraveling of the outer and inner DNA wrap of nucleosome respectively (Fig. 1d, left top panel), which is consistent with previous studies of ours[27,30,31] and others[32,33]. For the nucleosome with ubH2B (ubH2B-nucleosome), still two-step unfolding dynamics was observed, with force lower to around 2pN for the outer wrap and around 15pN for the inner wrap, as compared to that of H2B-nucleosome (Fig. 1d, left bottom panel). Statistical analysis on the measurements for hundreds of samples revealed that ubH2B-nucleosomes obtain a lower mechanical stability than H2B-nucleosomes (Fig. 1d, right panel). In addition, for both H2B and ubH2B-nucleosome, only the first stretching cycle showed the typical two-step unfolding pathway of intact nucleosome. In the following repeated stretching cycles, the two-step unfolding process of intact nucleosomes cannot be observed, which indicated that the nucleosome structure cannot be reassembled correctly after totally disrupted (Fig. 1e, with the 2D plots shown in Supplementary Fig. 1). This irreversibility might be due to the displacement of core histones from DNA when the nucleosome structure is fully unfolded[34]. Sometimes, we can observe an irregular one-step unfolding process in the second stretching cycle, which may be caused by some remaining histones bound on DNA that are displaced soon in the next stretching cycle. The results indicated that H2B mono-ubiquitination does not affect the folding and unfolding dynamic pattern of nucleosome, but impairs the stability of nucleosome mildly. To further quantify the effect of ubH2B on the stability of nucleosome, the free energy costs to unfold the outer DNA wrap of H2B- and ubH2B-nucleosome were calculated (please find more calculation details in Methods). Based on the distribution of rupture force, the free energy cost to unfold the outer wrap is evaluated to be ~40 kJ/mol and ~20 kJ/mol for H2B- and ubH2B-nucleosome respectively. Compared to H2B-nucleosome, a less energy cost to unfold the outer DNA wrap is caused by the mono-ubiquitination of H2B in nucleosome.

### ubH2B recruits FACT to stabilize and maintain the nucleosome state

To investigate how ubH2B affects the binding of FACT on nucleosomes, the recombinant FACT complex with the two subunits SPT16 and SSRP1 were purified and incubated with biotin-labelled unmodified H2B or ubH2B mono-nucleosomes (Supplementary Fig. 2a, b). The mono-nucleosomes pull-down assays showed that the purified FACT has a much higher affinity for the ubH2B-nucleosome (Fig. 2a), which indicated that H2B ubiquitination directly recruits FACT at the nucleosome level. Magnetic tweezers were further employed to investigate how ubH2B affects FACT's functions on nucleosome dynamics. In the presence of FACT complex, H2B-nucleosome was observed to be completely disrupted at tensions below 10pN, with the structural transitions reversible and retaining the similar force response in each repeated stretching cycle (Fig. 2b top panel). For H2B-nucleosome, FACT impairs the nucleosome stability greatly and helps to maintain the integrity of nucleosome after the nucleosome is fully

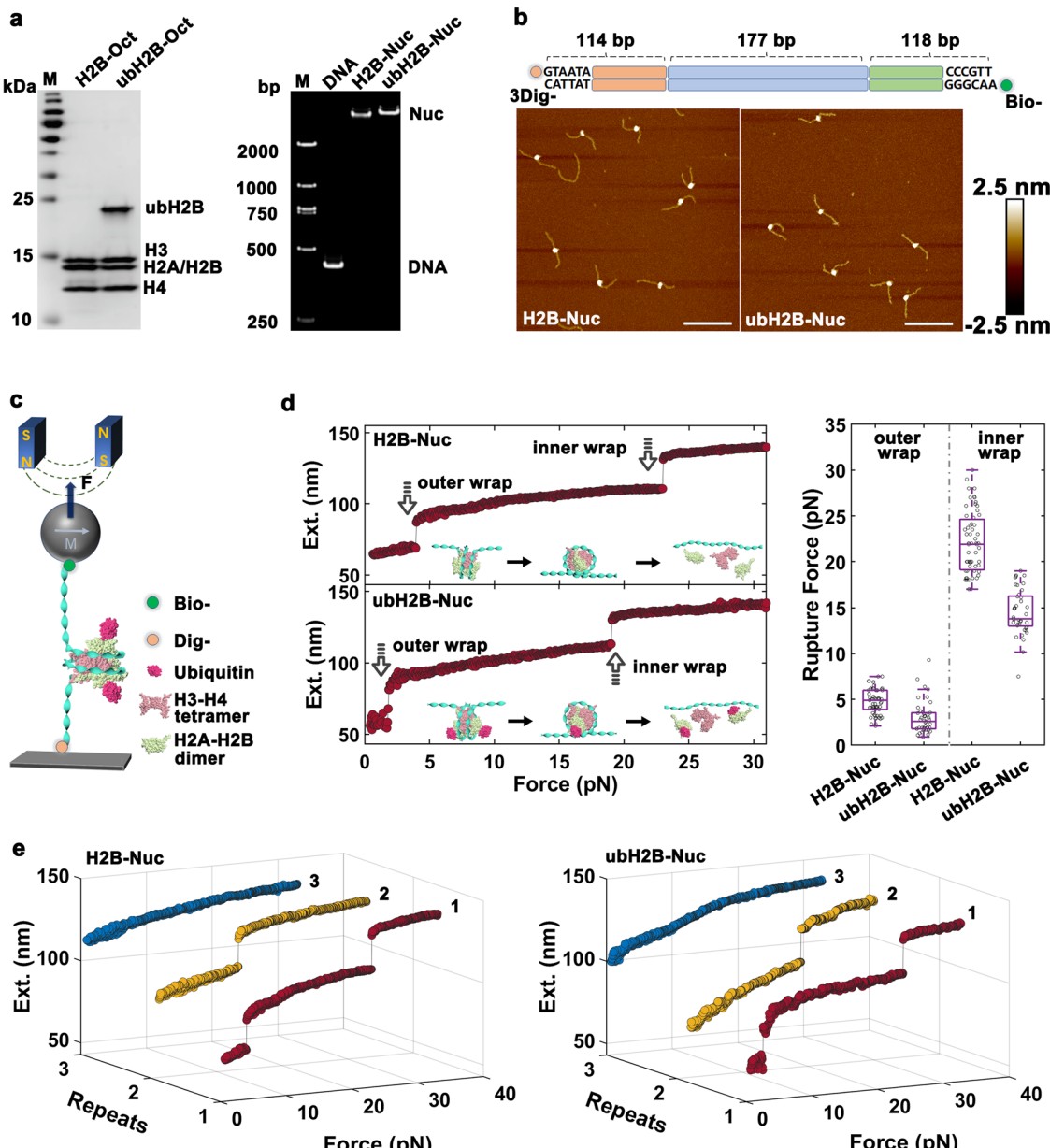

**Fig. 1 | ubH2B impairs the stability of nucleosome. a** SDS-PAGE analysis (left) of the purified histone octamers with unmodified H2B (H2B Oct) and ubH2B (ubH2B Oct), and 5% native acrylamide gel electrophoresis analysis (right) of reconstituted H2B-nucleosome (H2B-Nuc) and ubH2B-nucleosome (ubH2B-Nuc) for magnetic tweezers investigation, with free DNA template (DNA) as control. The data are representative of $n = 3$ biologically independent experiments. The first lane is the marker (M). **b** AFM images of unmodified H2B-(left) and ubH2B-nucleosome (right) reconstituted on the DNA template shown at top. Bar: 100 nm. Height: −2.5 nm to 2.5 nm. **c** A schematic representation of single-molecule stretching experiment by magnetic tweezers (not to scale). **d** The typical force-extension curves of unmodified H2B-nucleosome (top) and ubH2B-nucleosome (down) by magnetic tweezers, with the related statistic rupture forces for the outer and inner DNA wrap shown in right panel ($n = 115$ for H2B-nucleosome, $n = 53$ for ubH2B-nucleosome). The box plots include the median line (median value indicated), the box denotes the interquartile range (IQR), whiskers denote the rest of the data distribution, and outliers are denoted by points greater than $\pm 1.5 \times IQR$. **e** The typical repeated stretching measurements of H2B-nucleosome (left) and ubH2B-nucleosome (right). In each stretching cycle, the force is increased up to 32pN. Source data are provided as a Source Data file.

disrupted under tension, consistent with our previous results[27]. Intriguingly, for ubH2B-nucleosome, FACT does not impair, but greatly enhances the stability of ubH2B-nucleosome, and still maintains its integrity. As shown in Fig. 2b (bottom panel), in the presence of FACT complex, ubH2B-nucleosome was observed to unfold in a reversible two-step manner at much higher rupture forces of about 10pN for the outer wrap and around 20pN for the inner wrap. The repeated stretching measurements indicated that FACT can tether the histones to the DNA and facilitate the reassembly of ubH2B-nucleosome after the nucleosome is fully disrupted. As revealed by the statistical analysis

on hundreds of samples (Fig. 2c), FACT stabilizes the ubH2B-nucleosome and maintains the integrity of nucleosome. In addition, to investigate the direct binding capacity of FACT on H2B- and ubH2B-nucleosome, the dissociation contants $K_d$ for FACT binding to the 409-bp nucleosome were measured by single-molecule magnetic tweezers[27]. The resulting $K_d$ for FACT binding is $16.3 \pm 3.9$ nM (mean $\pm$ SE) for H2B-nucleosome and $8.0 \pm 4.2$ nM (mean $\pm$ SE) for ubH2B-nucleosome with significant difference $p < 0.01$ (please find more calculation details in Methods). The results showed that the purified FACT has a higher affinity for the ubH2B-nucleosome, which agree with the

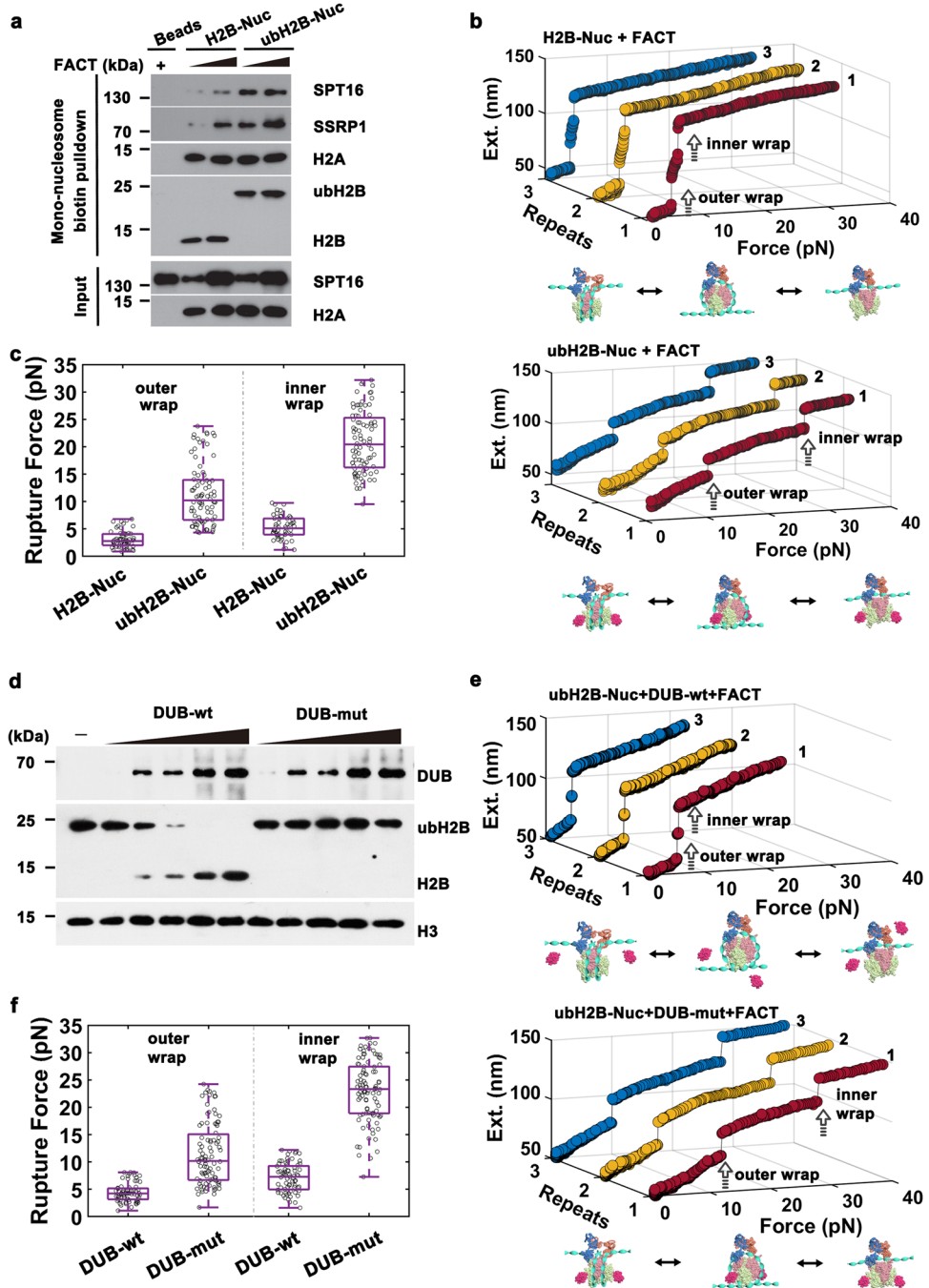

**Fig. 2 | ubH2B recruits FACT to stabilize and maintain nucleosome structure.**
**a** Mono-nucleosome pull-down assay of unmodified H2B-nucleosome (H2B-Nuc)
and ubH2B-nucleosome (ubH2B-Nuc) incubated with the purified FACT. The
immunoblots are representative of three biologically independent experiments
($n = 3$). **b** The typical repeated stretching measurements of unmodified H2B (top)
and ubH2B (bottom) -nucleosome with FACT. In each stretching cycle, the force is
increased up to 32 pN. **c** The statistic rupture forces for the outer and inner DNA
wrap for unmodified H2B-nucleosome and ubH2B-nucleosome incubated with
FACT ($n = 51$ for H2B-nucleosome, $n = 84$ for ubH2B-nucleosome). **d** In vitro deu-
biquitination assay for ubH2B nucleosome incubated with increasing amounts of
wild-type DUB (DUB-wt) and its catalytically inactive mutant (DUB-mut). Western
blot experiments show that ubH2B-nucleosomes are successfully deubiquitinated
by DUB-wt with high efficiency, but cannot be deubiquitinated by DUB-mut. The

data are representative of $n = 3$ biologically independent experiments. **e** The typical
repeated stretching measurements of ubH2B-nucleosome incubated with FACT
and DUB-wt (top) or DUB-mut (bottom). The nucleosome with ubH2B successfully
deubiquitinated by DUB-wt can be destabilized and maintained by FACT, while that
nucleosomes with ubH2B not deubiquitinated by DUB-mut are still stabilized and
maintained by FACT. **f** The statistic rupture forces for the outer and inner DNA wrap
for ubH2B-nucleosome incubated with FACT and DUB-wt or DUB-mut ($n = 69$ for
ubH2B-nucleosome with DUB-wt, $n = 95$ for ubH2B-nucleosome with DUB-mut).
The cartoons below represent the relative dynamic process of the nucleosome,
respectively. The box plots (**c**, **f**) include the median line (median value indicated),
the box denotes the interquartile range (IQR), whiskers denote the rest of the data
distribution, and outliers are denoted by points greater than ±1.5×IQR. Source data
are provided as a Source Data file.

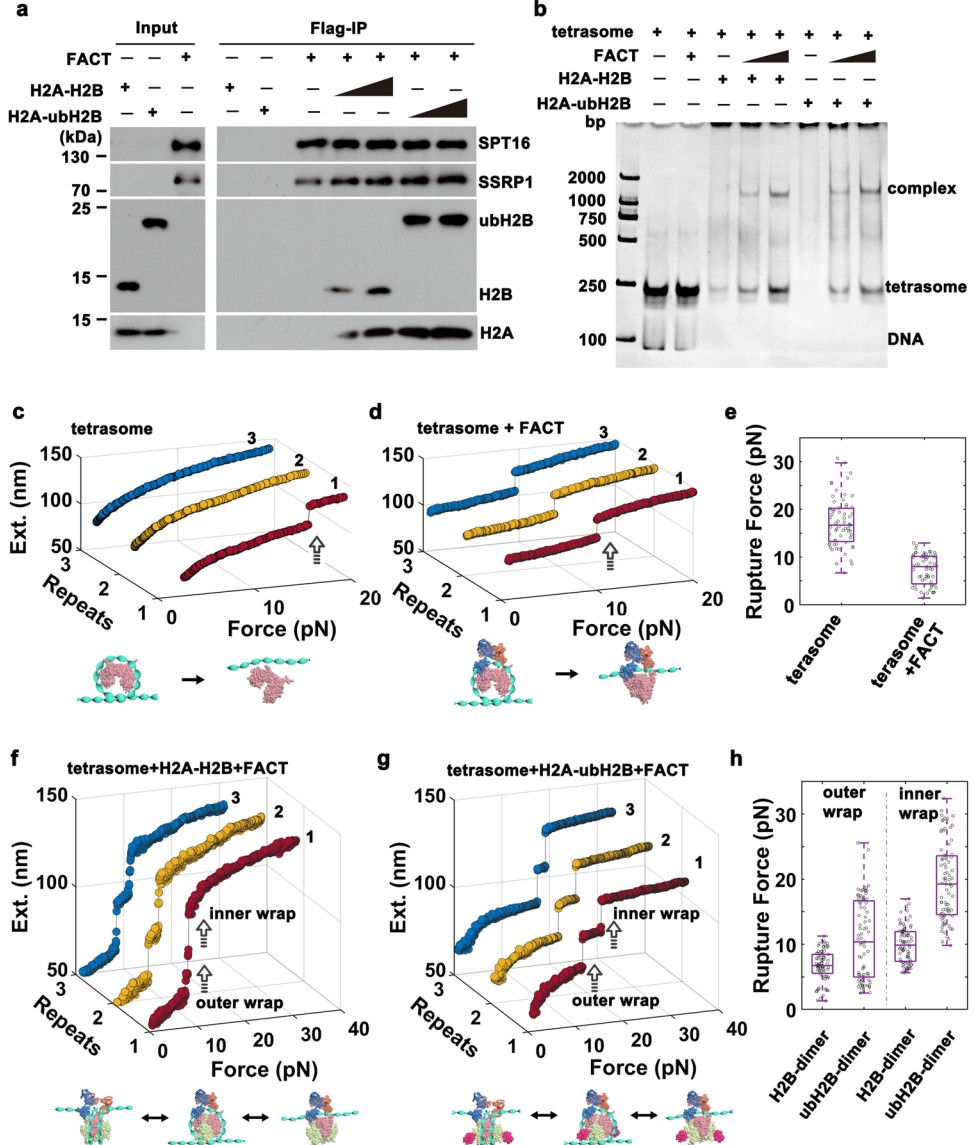

**Fig. 3 | FACT prefers to bind and deposit H2A-ubH2B dimer to form intact nucleosome. a** Immunoprecipitation (IP) assays of the recombinant FACT with Flag-SPT16 and SSRP1 incubated with recombinant H2A-H2B or H2A-ubH2B dimer. The results showed that FACT prefers to bind H2A-ubH2B dimer. The data are representative of $n = 3$ biologically independent experiments. **b** The gel electrophoresis analysis of the binding of FACT-H2A-H2B or FACT-H2A-ubH2B with (H3-H4)$_2$ tetrasome. The results showed that ubH2B does not affect the binding of FACT-H2A-H2B on (H3-H4)$_2$ tetrasome. The data are representative of $n = 3$ biologically independent experiments. The typical repeated stretching measurements of the mono-tetrasome alone (**c**) and that with FACT (**d**), with the statistical analysis of the rupture force shown ($n = 64$ for the tetrasome and $n = 59$ for the tetrasome with FACT) (**e**). The typical repeated stretching measurements of the mono-tetrasome with FACT and H2A-H2B dimer (**f**) and that with FACT and H2A-ubH2B dimer (**g**), with the statistical analysis of the rupture force shown ($n = 79$ for both H2A-H2B dimer and H2A-ubH2B dimer) (**h**). The box plots (**e** and **h**) include the median line (median value indicated), the box denotes the interquartile range (IQR), whiskers denote the rest of the data distribution, and outliers are denoted by points greater than±1.5×IQR. The cartoons below represent the relative dynamic process of the tetrasome or nucleosome, respectively. Source data are provided as a Source Data file.

mono-nucleosomes pull-down assays. In addition, previously investigations have shown that FACT prefers to bind the altered nucleosome structures, including partially unwrapped nucleosome, the sub-nucleosome, or the nucleosome with double-strand break site[26,35,36]. As we revealed that ubH2B can impair the stability of nucleosome, the allosteric effect of ubH2B on the nucleosome state may also function to recruit and correlate with FACT. Taken together, our results indicated that ubH2B recruits FACT to maintain a stabilized structural state at nucleosome level.

To further confirm that the stabilized effect of FACT on nucleosome is caused by ubH2B, we de-ubiquitinated the ubH2B on nucleosome by DUB protein and performed the same experiments. The in vitro de-ubiquitination assay showed that the ubH2B nucleosomes were successfully de-ubiquitinated by wild-type DUB (DUB-wt) in vitro with high efficiency, but not by its catalytically inactive DUB mutant (DUB-mut), as examined by western blot experiments (Fig. 2d). In addition, as shown in Supplementary Fig. 2c, d, the addition of DUB did not affect the unfolding dynamics of H2B nucleosome and the FACT's function on nucleosome. Force-extension measurements revealed that, after the ubH2B-nucleosome was de-ubiquitinated by DUB-wt, the nucleosome was disrupted at much lower force below 10 pN in the presence of FACT, with the tipical trace shown in the top panel of Fig. 2e. As compared, its enzymatic-inactive DUB-mut cannot restore the destabilized effect of FACT on nucleosome (Fig. 2e, bottom panel), with the statistical analysis shown in Fig. 2f and the related 2D plots shown in Supplementary Fig. 2e, h. These results clearly revealed that

the de-ubiquitination of ubH2B-nucleosome rescues the destabilized effect of FACT on nucleosome, which confirmed that H2B ubiquitination recruits FACT to maintain a stabilized structural state at nucleosome level.

### FACT prefers to bind and deposit H2A/ubH2B dimer to form intact nucleosome

As a chaperone of histone H2A/H2B dimer, the FACT complex has been found to deposit H2A/H2B dimer onto the $(H3/H4)_2$ tetrasome (the particles made of DNA wrapped around an H3/H4 tetramer) to form intact nucleosome[19,27]. We have found that ubH2B recruits FACT to stabilize and maintain the nucleosome state, then how does ubH2B regulate the chaperone property of FACT? We first examined the effect of ubH2B on the binding property of FACT with H2A-H2B dimer. The recombinant H2A-H2B dimer or H2A-ubH2B dimers were incubated with the purified recombinant FACT complex, composed of Flag-SPT16 and SSRP1. As shown by the immunoprecipitation assays (Fig. 3a), FACT prefers to bind H2A-ubH2B dimer. We then examined the effect of ubH2B on the binding of FACT-H2A-H2B on $(H3-H4)_2$ tetrasome. The gel electrophoresis analysis showed that FACT can promote the binding of H2A-H2B to tetrasomes to form a ternary complex with FACT-H2A-H2B and $(H3-H4)_2$ tetrasome, and ubH2B does not have a pronounced effect on the binding of FACT-H2A-H2B on $(H3-H4)_2$ tetrasome (Fig. 3b).

The effect of ubH2B on the property of FACT-mediated nucleosome reassembly was examined by magnetic tweezers. The mono-$(H3-H4)_2$ tetrasomes were reconstituted in vitro onto the 409 bp DNA fragment with one Widom 601 nucleosome positioning sequence, as characterized by AFM imaging (Supplementary Fig. 3a, b). The mechanical property of mono-$(H3-H4)_2$ tetrasomes was first investigated through the stretching measurements. The typically irreversible one-step transition at force around 16 pN corresponds to the disruption of the inner nucleosome wrap (Fig. 3c), which is consistent with previous studies[27]. In the presence of FACT, a reversible one-step transition at lower force around 7 pN indicated that FACT impairs the stability of tetrasome and maintains its integrity (Fig. 3d), with the statistical analysis shown in Fig. 3e. For the tetrasomes incubated with both FACT and H2A-H2B dimer, a reversible two-step structural transition was observed at force less than 10 pN in the repeated stretching cycles (Fig. 3f), which indicates that FACT can deposit the H2A-H2B dimer onto tetrasome to form an intact nucleosome. To further confirm that two H2A-H2B dimer were deposited onto the tetrasome to form an intact nucleosome, we conducted the in situ deposition experiment by magnetic tweezers (Supplementary Fig. 3c, d). Obviously two sequential 10-nm deposition steps were observed, which corresponds to the sequential deposition of two H2A-H2B dimers. After the real-time deposition process, we carried out the force-extension measurement for the same molecule, and a reversible two-step structural transition of intact nucleosome were observed. Interestingly, in the presence of both FACT and H2A-ubH2B dimer as shown in Fig. 3g, still an obviously reversible two-step unfolding pathway was observed, but the rupture forces were distributed up to around 10 pN and 20 pN (with the related 2D plots shown in Supplementary Fig. 3e–h), similar to the mechanical property of ubH2B-nucleosome with FACT (Fig. 2b, c). The statistical analysis (Fig. 3h) and the in situ deposition experiment (Supplementary Fig. 3d) further confirmed that FACT deposits the H2A-ubH2B dimer onto tetrasome to form intact ubH2B-nucleosome. After the deposition process, FACT remains to bind the formed ubH2B-nucleosome to stabilize and maintain the nucleosome state. The effect of ubH2B on the deposition ability of FACT for H2A-H2B dimer was further quantitatively analyzed by magnetic tweezers. After the incubation of tethered tetrasomes with FACT and H2A-H2B or H2A-ubH2B mixture for 30 minutes, $79.1 \pm 3.5\%$ of the tetrasomes were successfully assembled into nucleosome for H2A-ubH2B dimer, as compared to that of $54.5 \pm 5.3\%$

for H2A-H2B dimer (Supplementary Fig. 3i). As a conclusion, ubH2B enhances the chaperone property of FACT; FACT prefers to bind H2A-ubH2B dimer and helps to deposit the H2A-ubH2B onto tetrasome to form intact nucleosome.

### ubH2B recruits FACT to activate gene transcription

Our in vitro investigation showed that the ubH2B helps to recruit FACT to form a stable and robust structural state at the nucleosome level. We then explored how the ubH2B cooperates with FACT to regulate gene expression in mouse embryonic stem cells (mESCs). The genome-wide correlation between the levels of ubH2B and SSRP1 (one subunit of FACT complex) by chromatin immunoprecipitation (ChIP)-seq analysis were examined as shown in Fig. 4a−c and Supplementary Fig. 4a, with their distribution on specific gene regions shown in Fig. 4d. A total of 9786 SSRP1 and 16,016 ubH2B enrichment sites were identified by model-based analysis of ChIP-seq peak calling (Fig. 4a). Among them, 2489 SSRP1 peaks overlapped with 2344 ubH2B peaks (overlapped region in Fig. 4a). For all the genes enriched with SSRP1, the read density of SSRP1 for the genes with SSRP1 only or those with both SSRP1 & ubH2B overlapped were analyzed (Fig. 4b). The results reveal that more SSRP1 are enriched in the gene regions containing ubH2B, as compared to those without ubH2B, which indicated that ubH2B helps to recruit more FACT on chromatin. In addition, we analyzed the expression level between the genes enriched with SSRP1 only, both SSRP1 & ubH2B overlapped, and SSRP1 desert which has no SSRP1 ChIP-seq peak (Fig. 4c for whole genome and Fig. 4d for specific and typical gene regions). The results showed that FACT facilitates gene transcription, with SSRP1 containing genes, no matter with or without ubH2B, significantly more active than SSRP1 desert genes. Significantly, we observed that ubH2B correlates with FACT to facilitate gene transcription, with ubH2B & SSRP1 overlapped genes significantly more active than the SSRP1 only genes (Fig. 4c) and ubH2B only genes (Supplementary Fig. 4a). The genome-wide analysis showed that ubH2B helps to recruit more FACT to facilitate gene transcription.

To further explore the dynamic interplay of ubH2B and FACT on the transcriptional process, time-course ChIP-qPCR analysis was employed on the gene body regions of RAR-RXR regulated genes during transcriptional activation by all-trans retinoic acid (tRA) induction. The results for the typical examples of genes *Phrf1*, *Sipa1*, *Erf* and *Sxrn1* were shown in Fig. 4e, f and Supplementary Fig. 4b, c. The mRNA expression level of the genes was increased with tRA treatment in a time-dependent manner, as shown in the left panels of Fig. 4e, f and Supplementary Fig. 4b, c. Interestingly, during the activation process of transcription, the rapid increase of ubH2B was observed to be accompanied by gradually increase of SSRP1 on the gene body regions, with the level of total H2B not changed (Fig. 4e, f and Supplementary Fig. 4b, c, right panel). The results indicated that during the transcriptional elongation process, the nucleosome level is maintained in the gene region, with H2B quickly ubiquitinated before the recruitment of FACT, which is consistent with our in vitro analysis. Taken together, our genome-wide study and the time-course ChIP-qPCR analysis on specific genes confirmed that ubH2B helps to recruit FACT to facilitate gene transcription.

## Discussion

The mono-ubiquitination of histone H2B at lysine 120 (ubH2B) has long been revealed to play an important role in regulating genome functions, especially in transcriptional process. Previous investigations on the roles of ubH2B mainly rely on the manipulation of the major H2B ubiquitination ligase BRE1 in yeast or RNF 20/40 in mammalian cells[6,7], while how ubH2B directly regulates the dynamics of nucleosome and functions in transcriptional process remain largely unknown. In this study, we intensively discovered the direct effect of ubH2B on nucleosome dynamic and its correlation with FACT complex at

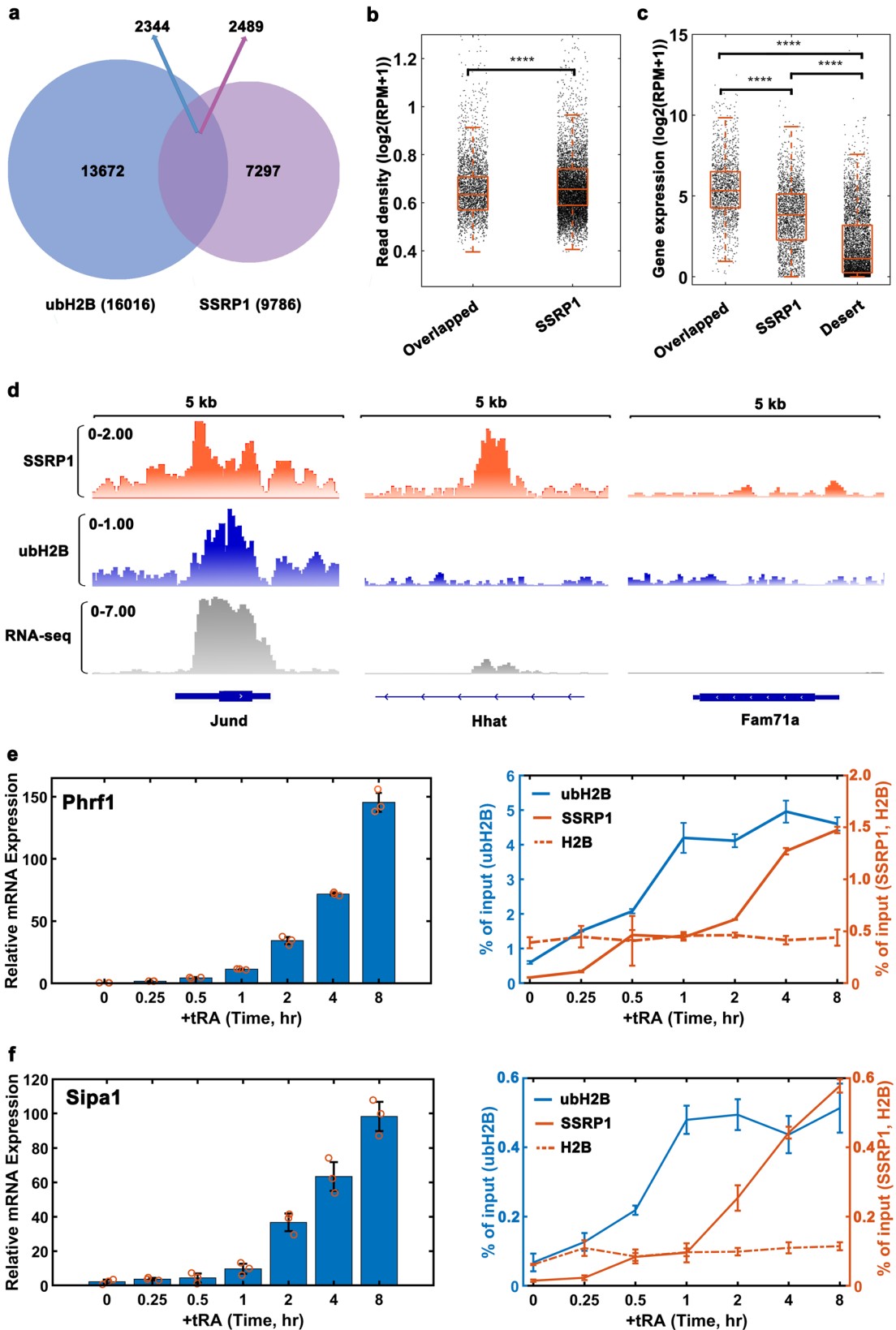

nucleosome level (Fig. 5). We found that ubH2B itself does not affect the folding and unfolding pattern of nucleosome, but impairs the stability of nucleosome. The nucleosome with ubH2B maintains the typically irreversible two-step unfolding dynamics, but unfolds at lower forces to around 2pN for the outer wrap and around 15pN for the inner wrap, as compared to that for unmodified H2B-nucleosome. The

results are consistent with previous in vitro analyses which found that ubH2B mildly destabilizes the Nap1-assisted nucleosome stability[15]. In addition, we further analyzed the free energy cost to unfold the outer wrap of ubH2B-nucleosome, and found that compared to H2B-nucleosome, a less energy cost of ~20 kJ/mol is caused by the ubiquitination of H2B in nucleosome.

**Fig. 4 | ubH2B recruit FACT to activate gene transcription by genome-wide and specific gene analysis. a** Venn diagram showing the overlap of ubH2B (blue) and SSRP1 (purple) peaks in mESCs. A total of 2489 SSRP1 peaks overlapped with 2344 ubH2B peaks. **b** Genome-wide analysis on the read density of SSRP1 for the genes enriched with SSRP1 only and those with SSRP1/ubH2B overlapped. ****Significant difference ($p < 0.0001$, with $p = 4.5E-9$; $n = 2489$ for Overlapped, $n = 7297$ for SSRP1) according to the two-sided student test. **c** Genome-wide analysis on the expression level of genes enriched with SSRP1/ubH2B overlapped, SSRP1 only and SSRP1 desert (the genes without SSRP1 ChIP-seq peak). ****Significant difference ($p < 0.0001$, with $p = 1.0E-151$, $5.6E-281$ and $0$ for the comparision of Overlapped and SSRP1, SSRP1 and Desert, Overlapped and Desert, respectively; $n = 1638$ for Overlapped, $n = 2304$ for SSRP1 and $n = 5000$ for Desert) according to the two-sided student test. The box plots (**b**, **c**) include the median line (median value indicated), the box denotes the interquartile range (IQR), whiskers denote the rest of the data

distribution, and outliers are denoted by points greater than ±1.5×IQR. **d** The distribution of SSRP1 and ubH2B at indicated genes as well as their corresponding expression levels, for genes enriched for SSRP1/ubH2B overlapped (left column), SSRP1 only (middle column), and SSRP1 desert (right column). The y-axis is the read density (log2(RPM+1)) from the ChIP-seq data for SSRP1 and ubH2B, and gene expression (log2(RPM+1)) from RNA-seq data, respectively. RPM: the reads per million mapped reads. The chromatin regions enriched for both FACT and ubH2B are associated with highly transcribed genes. ChIP-qPCR analysis (right) of the level of SSRP1, ubH2B and H2B on the gene body regions of Phrf1 (**e**) and Sipa1 (**f**), with the relative levels of mRNA shown (left) at different time points during tRA induction as measured using RT-real time-PCR. The levels were normalized as n-fold changes relative to the values prior to tRA induction. All the data represent means ± S.D. (standard deviation) of three independent biological replicates ($n = 3$). Source data are provided as a Source Data file.

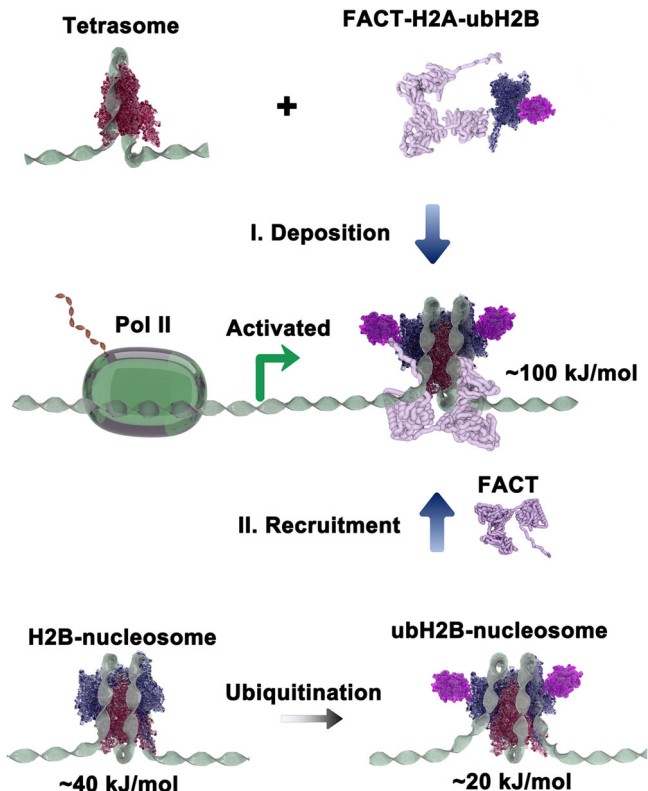

**Fig. 5 | Mechanism model proposed for the co-regulation of ubH2B and FACT on gene transcription.** The preferable binding of FACT on ubH2B-nucleosome forms a stable altered nucleosome state and provides a key platform for transcriptional activation. I. ubH2B promotes the binding of FACT to deposit the H2A-ubH2B dimer to form a stable FACT-ubH2B-nucleosome state. II. Ubiquitination of H2B impairs the stability of nucleosome, but helps to recruit FACT to maintain a stable structural state at nucleosome level.

Our and previous in vitro investigations on the direct effect of ubH2B on nucleosome dynamic revealed that ubH2B impairs the stability of nucleosome, especially the strength of interactions between the DNA and histones. However, previously investigations in yeast cells have showed that ubH2B is required for efficient reassembly of nucleosomes during Pol II-mediated transcription elongation, enhance the nucleosome stability, and mediate both activation and repression of transcription depending the localization and context of the modification[16,18]. How can ubH2B fulfill these functions in cells? Our research suggests that although the ubH2B itself cannot enhance the nucleosome stability and help nucleosome reassembly, but it can act as a structural platform to recruit FACT complex to fulfill these functions. We showed that ubH2B can help to recruit FACT by enhancing

the binding of FACT on nucleosome, and the binding of FACT on ubH2B-nucleosome greatly enhances the nucleosome stability. The free energy cost for the outer wrap of ubH2B-nucleosome in the presence of FACT is increased up to ~100 kJ/mol. At the same time, FACT helps to maintain the ubH2B-nucleosome integrity. In addition, ubH2B promotes the binding of FACT on H2A/H2B dimer, maintains and enhances the FACT's chaperone function to deposit H2A/H2B dimer to form intact nucleosome. Our genome-wide analysis and time-course ChIP-qPCR analysis on specific genes also indicated that ubH2B helps to recruit more FACT to facilitate gene transcription. As a structural platform, ubH2B recruits FACT complex to help to maintain the integrity of ubH2B-nucleosome and form a stable and altered nucleosome state, which is ready for transcriptional activation (Fig. 5). Previously, single-molecule DNA unzipping invesigations showed that ubH2B increased the overall barrier strength of Pol II transcription through a nucleosome[17]. It is of great interest to investigate how ubH2B correlated with Pol II and the relevant proteins during the transcriptional process. In addition, ubH2B has been found to stimulate H3 methylation on lysine 4 (H3K4) and on lysine 79 (H3K79) by recruiting histone methyltransferase Set1/COMPASS and Dot1 respectively, and has been found to associate with the SWI/SNF chromatin remodeling complex for transcriptional activation[37–39]. Structural analyses on the ubH2B-nucleosome with COMPASS or Dot1L revealed that the ubiquitin modified on H2B serves as a structural platform to recruit and contact with the proteins for functional stimulation[40–47]. Other than the direct effect of ubH2B on the property of chromatin, the indirect effect of ubH2B to act as a structural platform by recruiting or preventing the binding of some specific proteins, might function more importantly to regulate the transcriptional process. In addition to its role in gene transcription, ubH2B has also been found to play roles in the processes of DNA replication and DNA damage response[48,49]. Our investigation on the directly mechanism of ubH2B's functions in the nucleosome dynamic and in its correlation with FACT may help to understand these related processes.

The histone chaperone FACT has been found to play essential roles in nucleosome remodeling during the chromatin-related processes, including DNA transcription, replication and repair. In these processes, the nucleosome barrier needs be temporarily displaced to expose DNA for processing, and then be rapidly reassembled to protect the DNA and preserve the original epigenetic identity[3,4]. Previous studies have found that FACT appears to function not only in destabilizing nucleosome to facilitate the progression of DNA and RNA polymerases on chromatins, but also in establishing and maintaining the genome-wide integrity of chromatin structure[23–25]. Most recently, we and others have revealed that FACT can tether both the (H3-H4)$_2$ tetramer and H2A-H2B dimer on DNA by its two subunits SPT16 and SSRP1, which function distinctly but coordinately to fulfill the apparently opposite functions of FACT on nucleosome by in vitro structural and single-molecular investigations[26,27]. It is of great interest to

investigate how the FACT's functions are regulated by different epigenetic factors, as for histone mono-ubiquitination. Our previous study has revealed that the direct regulation of H2A ubiquitination on the two-face functions of FACT complex at nucleosome level. ubH2A has been found to inhibit the binding of FACT on nucleosome to block the FACT's function at nucleosome level, but not affect the FACT's chaperone function in nucleosome assembly. At this point, FACT functions as a "salvager" to maintain ubH2A-nucleosome as a repressed state of genes, by helping to reassemble the ubH2A-H2B to form intact ubH2A-nucleosome, but cannot disassemble the ubH2A-nucleosome. Previous analysis by in vitro transcription assays showed that ubH2B can cooperate with the FACT complex to stimulate the efficiency of RNA Pol II elongation, and suggested that it may caused by the efficient displacement of H2A/H2B dimer from the nucleosome barrier[29]. However, our purified system in this work clearly showed that although ubH2B does not affect the FACT's chaperone function in nucleosome assembly, it helps to recruit FACT and greatly impairs the destabilized effect of FACT on nucleosome state. The preferable binding of FACT on ubH2B-nulceosome helps to maintain a stable and altered nucleosome state for transcriptional activation.

The same "ubiquitin" modified at different sites on nucleosome has totally different effect on FACT's function. Recent cryo-EM structural analysis on the FACT complex with partially assembled sub-nucleosome has revealed that the dimerization domains of SPT16 and SSRP1 straddle on the dyad of nucleosome with the middle domains of SPT16 and SSRP1 projecting down on the either side of nucleosome's face, as like a unicycle[26]. The ubiquitin modified at H2A-K119 site around the dyad of nucleosome may clash and exclude the binding of FACT on nucleosome; while the ubiquitin modified at H2B-K120 site on the face of nucleosome may contact with the middle domains of SPT16 and SSRP1 to recruit and stabilize the binding of FACT on nucleosome. In addition, FACT has been found to prefer to bind the partially unwrapped nucleosome[36], the sub-nucleosome[26], or the nucleosome with double-strand break site[35], rather than the intact nucleosome. Our maganeitc tweezer investigations have revealed that ubH2A greatly enhances, but ubH2B impairs the stability of nucleosome, which indicate that the allosteric effect of ubH2A and ubH2B on the nucleosome state may also function to correlate with FACT. It will be of great interest to resolve the structure of FACT with ubH2B-nucleosome to give more details on the regulation. The FACT complex, functioned as an essential and conserved factor in almost all chromatin-based processes, including DNA transcription, replication and repair in eukaryotes. It is critically important to understand how the core functions of FACT are regulated in different chromatin circumstances. Our research on the distinct regulation of ubH2B on FACT's functions provides a mechanistic insight to understand the FACT's function. It will be of great interest to investigate the regulation of FACT's function by other related epigenetic factors, which may help to propose a unifying model for FACT's activity at the chromatin state.

## Methods

### Protein purification
Histones H2A, H2B, H3 and H4 were cloned into pET3a histone expression plasmid, overexpressed in *E. coli*, and purified from inclusion bodies[50]. The purified histone ubH2B was purchased from KS-V Peptide Biological Technology Co., Ltd (Hefei, China), by total chemical synthesis utilizing standard solid phase peptide synthesis and peptide fragments ligation reactions to generate the desired protein[51]. For the purification of recombinant FACT complex[19,27], Sf9 cells ($1.5-2 \times 10^6$/ml) were infected with baculovirus containing Flag-SPT16 and His$_6$-SSRP1, and incubated at 27 °C for 72 h. The infected cells were collected and washed with ice-cold PBS, and then lysed in lysis buffer (150 mM NaCl, 20 mM Tris-HCl, pH 8.0, 5% glycerol, 1 mM PMSF). 10 mg (60 mL) of the cell extracts were incubated with 500 μL anti-Flag

M2-agarose (Sigma, A2220) for 4 h at 4 °C, with the resin washed by lysis buffer. Bound proteins were eluted in the presence of 0.5 mg/ml Flag peptide (Sigma, F4799) and further purified by a Heparin HP column (GE Healthcare). The fractions containing FACT complex were dialyzed against BC-100 buffer (100 mM NaCl, 10 mM Tris-HCl, pH 8.0, 0.5 mM EDTA, 20% glycerol, 1 mM DTT, 1 mM PMSF) and stored at −80 °C.

### Nucleosome or tetrasome reconstitution
To reconstitute the respective histone octamers, H3/H4 tetramers and H2A/H2B dimers[27], equimolar amounts of individual histones in unfolding buffer (7 M guanidinium HCl, 20 mM Tris-HCl, pH 7.5, 10 mM DTT) were dialyzed into refolding buffer (2 M NaCl, 10 mM Tris-HCl, pH 7.5, 1 mM EDTA, 5 mM 2-mercaptoethanol), and purified by a Superdex S200 column. For magnetic tweezers' investigation, a 409 bp DNA template with a single 601 sequence in the middle was prepared by PCR from plasmid using a biotin(bio)-labeled forward primer and a 3×digoxigenin (3×dig)-labeled reverse primer. Chromatin samples were assembled using the salt-dialysis method[50]. The reconstitution reaction mixture with octamers and 601 based DNA templates in TEN buffers (10 mM Tris-HCl, pH 8.0, 1 mM EDTA, 2 M NaCl) were dialyzed over 16 h at 4 °C in TEN buffer, which was continuously diluted by slowly pumping in TE buffer (10 mM Tris-HCl, pH 8.0, 1 mM EDTA) to a lower concentration of NaCl from 2 M to 0.6 M, followed by a final dialysis step in TE buffer for 4 h. The stoichiometry of octamer/tetramer to DNA template was determined by gel electrophoresis analysis and AFM[31,50].

### In vitro histone deubiquitination assays
1 μg ubH2B-nucleosomes were incubated with increasing concentrations (2.5 nM, 5 nM, 7.5 nM, 10 nM and 12.5 nM) of wild-type DUB (DUB-wt, recombinant USP21) and its catalytically inactive mutant (DUB-mut, USP21-C221A) in deubiquitination reaction buffer (50 mM HEPES pH 7.6, 150 mM NaCl, and 5 mM DTT) at 30 °C for 1 hr. The reaction was terminated by the addition of SDS−PAGE sample loading buffer. The proteins were then resolved on SDS−PAGE and blotted with the anti-H2B antibody (Abcam, ab1790, 1:2000) and anti-His antibody (Huaxingbio, HX1804, 1:2000) for his-tagged DUB, with the uncropped and unprocessed scans of all of the blots shown in the Source Data file.

### Mono-nucleosome Pull-down Assay
Biotin-labeled DNA was prepared by PCR with biotin-labeled primer. 1 μg Biotin-labeled mono-nucleosomes and streptavidin-agarose-beads (10 μL) were mixed in BC300 buffer (20 mM Tris-HCl, pH 7.5, 10 %glycerol, 300 mM NaCl, 0.1%NP-40, 500 μg/mL BSA), and incubated with increasing concentrations (4 nM and 8 nM) of FACT complexes overnight at 4 °C[28]. The beads were washed by wash buffer (20 mM Tris-HCl, pH 7.5, 10 % glycerol, 300 mM NaCl, 0.5% NP-40) for 3 times, 10 min each time. The proteins were eluted with SDS-loading buffer (50 mM Tris, pH 6.8, 2%SDS, 0.1%Bromophenol blue, 10 %Glycerol, 1%beta-mercaptoethanol) and analyzed by western blot, with the uncropped and unprocessed scans of all of the blots shown in the Source Data file.

### Flag-IP assay
10 μL anti-Flag M2-agarose (Sigma, A2220) beads for each reaction was washed by PBS twice and pre-cleaned in BC300 buffer for 1 hr at 4 °C. Then Flag-FACT(8 nM) and increasing concentrations (1.5 nM and 3 nM) of H2A-H2B dimer or H2A-ubH2B dimer were added and incubated overnight at 4 °C[28]. The beads were washed 3 times with the wash buffer (20 mM Tris-HCl, pH 7.5, 10 % glycerol, 300 mM NaCl, 0.5 % NP-40), 10 min each time. The proteins were eluted by SDS-loading buffer and analyzed by western blot, with the uncropped and unprocessed scans of all of the blots shown in the Source Data file.

## Tetrasome binding assays

Tetrasomes for histone deposition assays were reconstituted on 87 bp DNA fragments using the salt-dialysis method. Different concentrations (25 nM and 50 nM) of FACT complexes were first incubated with H2A/H2B dimer (100 nM) at 4 °C for 30 min, and then mixed with the constant reconstituted tetrasomes (100 nM) in reaction buffer (10 mM HEPES, pH 8.0, 1 mM EDTA, 60 mM NaCl). The samples were incubated at 30 °C for 1 hr prior to electrophoresis on 5% native PAGE electrophoresis in 0.25 × TBE buffer (22.5 mM Tris, pH 8.0, 22.5 mM boric acid, 0.5 mM EDTA) for 1.5 hr at 80 V. The gels stained with ethidium bromide were scanned by a Gel Doc™ EZ Imager system (Bio-Rad), with the uncropped and unprocessed scans of all of the blots shown in the Source Data file.

## AFM analysis

The reconstituted nucleosome samples (20 ng/μL) were prepared in HE buffer and fixed with 0.1%glutaraldehyde (Fluka) on ice for 30 min. Rinse the mixture with 500 μL HE buffer for 2−3 times in a vivaspin500 column. The column was spined at 15 000 g for 2−5 min at 4 °C to get rid of the liquids and diluted to 0.5−1 ng/μL in HE buffer. 20 μL spermidine (1 mM) was dropped onto newly cleaved mica surface and incubated for 10 min. Then rinse the mica with 200 μL ddH$_2$O for 4 times and blow to dry mica surface briefly with nitrogen gas. 10 μL sample solution was added onto the mica surface, incubated for 10−15 min. Finally, wash the mica with 200 μL ddH$_2$O for 3 times and blow to dry gently. The prepared AFM samples were examined using ScanAsyst Mode of AFM (MultiMode 8 SPM system, BRUKER).

## Single-molecule magnetic tweezers analysis

The single-molecule stretching experiments were performed by magnetic tweezers[31]. Briefly, the two DNA ends of reconstituted nucleosome were tethered by digoxigenin and anti-digoxigenin ligation to a glass coverslip and by biotin-streptavidin ligation to a 2.8-μm diameter Dynabeads (M280, Invitrogen Norway), respectively. The applied tensions on the super paramagnetic beads arised from the strong magnetic field gradient of the two small NdFeB magnets, which were controlled to pull on the Dynabeads and thus stretch the DNA molecule on nucleosome. The bead image was projected onto the CCD camera (MC1362, Mikrotron) through an inverted microscope objective (UPLXAPO60XO, NA 1.42, Olympus). The real-time position (x, y, z) of the bead at various forces was recorded by comparing the diffraction pattern of the bead with calibration images at various distances from the focal point of the objective. The quadrant-interpolation (QI) algorithm was applied by LabView software to trace the three-dimensional position of the beads in the flow cell[52]. About 80 samples were traced simultaneously in each measurement. In the repeated stretching cycle, magnets were moved near to the samples at a rate of 10 μm/s, during which the dynamic unfolding of the nucleosome was recorded. After that, force was reduced to 0.1pN rapidly and waited for 5 min, and then repeated the stretching cycle on the same nucleosome. All measurements were carried out at 25 °C. To calibrate the force in the force-extension measurement, the force calculation for a 10,000 bp DNA tethered between the bead (M280) and the coverslip was carried out. At each magnet position, the y-position of the bead was recorded at 500 Hz for 5 mins and the corresponding force was calibrated by power-spectral-density (PSD) analysis[53]. The force measurements were repeated for 10 independent DNA tethers. The relationship between force and magnet position was fitted well with a double exponential function. Forces in force-extension measurements of nucleosomes were derived from the fitted double exponential function with magnets positions.

## Free energy calculation for the outer DNA wrap of nucleosome

Force-dependent folding free energy $\triangle G(f)$ of outer DNA wrap of nucleosome were determined by force-dependent unfolding rate $k_{f-u}(f)$ and folding rate $k_{u-f}(f)$:

$$\triangle G(f) = k_B T \ln \frac{k_{u-f}(f)}{k_{f-u}(f)} \tag{1}$$

The kinetic rate constants $k_{f-u}(f)$ and $k_{u-f}(f)$ were derived from the distribution of the lifetime at constant force $f$. When $\Delta G(f) = 0$, we read the equilibrium force $F_{eq}$ from the point at which $\ln K = \ln \frac{k_{u-f}}{k_{f-u}} = 0$.

$\triangle G(f)$ induces a force-independent term $\triangle G_0$ and a force-dependent term $\triangle\Phi(f)$:

$$\triangle G(f) = \triangle G_0 + \triangle\Phi(f) \tag{2}$$

where

$$\Delta\Phi(f) = -\int_0^f \left( x_{dna}(f') - x_{ncp}(f') \right) df', \tag{3}$$

$x_{dna}(f)$ is the extension of the unfolded DNA chain at force $f$, $x_{ncp}(f)$ is the extension of the folded nucleosome at force $f$[54]. $x_{dna}(f)$ and $x_{ncp}(f)$ has been shown to follow the worm-like chain (WLC) polymer model and can be obtained by numerical inversion of equation:

$$\frac{fA}{k_B T} = \frac{1}{4(1 - x_{dna}/L)^2} - \frac{1}{4} + \frac{x_{dna}}{L}, \tag{4}$$

where persistence length of $A = 50$ nm[55] and contour length of $L$.

When $\Delta G(F_{eq}) = 0$, from Eq. (2), the free energy

$$\Delta G_0 = -\Delta\Phi(F_{eq}) = \int_0^{F_{eq}} \left( x_{dna}(f') - x_{ncp}(f') \right) df'. \tag{5}$$

For a molecule of H2B-nucleosome as shown in Supplementary Fig. 5a, $L$ is 119 nm for $x_{dna}(f)$ and 99 nm for $x_{ncp}(f)$. The equilibrium force is read directly $F_{eq}$=3.5 pN from Supplementary Fig. 5a right panel. According to Eq. (5), the free energy was calculated $\triangle G_0 = 38.8$kJ/mol.

Based on the kinetic rate analysis under tensions, the equilibrium force $F_{eq}$ can be determined precisely for each molecules. However, to build the distribution of the free energy by collecting adequate samples with well two-state transition behavior is a great challenge. For simplicity, we chose the media value of the rupture force for the equilibrium force based on the force distribution of the outer DNA wrap. For H2B-nucleosome, we choose 4.0pN for the equilibrium force based the distribution as shown in Fig. 1d right panel. The corresponding free energy cost was calculated to be ~40 kJ/mol which is consistent with the value derived from the kinetic rate analysis. For ubH2B-nucleosome, we choose 2.1pN for the equilibrium force and the free energy cost was calculated to be ~20 kJ/mol. For ubH2B-nucleosome combined with FACT, we choose 10.1 pN for the equilibrium force based on the distribution as shown in Fig. 2c. The corresponding free energy cost was calculated to be ~100 kJ/mol.

## The dissociation constant $K_d$ measurement by single-molecule magnetic tweezers

We performed the $K_d$ measurement for FACT binding to the 409 bp nucleosome by single-molecue magnetic tweezers[27].

In the case: nucleosome + FACT $\rightleftharpoons$ complex, the value $K_d$ is defined as:

$$K_d = \frac{[nucleosome][FACT]}{[complex]} = \frac{[nucleoosme]}{[complex]}[FACT] = \frac{N_{nucleosome}}{N_{complex}}[FACT]. \tag{6}$$

In the flow cell of magnetic tweezers, the number of FACT anchored to nucleosome on the surface of coverslip is much lower

than the number of FACT in the flow cell. We can ignore the change of concentration of FACT in the flow cell caused by nucleosome binding. The $K_d$ value could be obtained by calculating the ratio of the number of nucleosome alone to the number of complex at a certain concentration of FACT. The force-extension measurement has revealed that canonical nucleosome are totally disrupted at the tension around 25pN in the absence of FACT, and at the tension less than 10pN in the presence of FACT. When we exerted 11pN on the nucleosome sample, the inner wrap of nucleosome without FACT will not be disrupted while the inner wrap of the nucleome bound with FACT will be disrupted (Supplementary Fig. 5b). The different mechanical response of nucleome at force 11pN can distinguish whether FACT binds to nucleosome. The force-jump experiments were performed between 0.1pN and 11pN tension for tethered nucleosome samples for hundreds of stretching cycles, with the state number of nucleosome without FACT ($N_{nucleosome}$) and of nucleosome combined with FACT ($N_{complex}$) counted at 83 nM FACT concentration. The similar measurements of the $K_d$ between FACT and ubH2B-nucleosome were performed. The outer wrap of the ubH2B-nucleosome disrupted at ~3pN in the abence of FACT, and disrupted at the tension around 10pN in the presence of FACT. When the force of 6pN were exterted on the ubH2B-nucleosome sample, the outer wrap of ubH2B-nucleosome without FACT will be disrupted while the nucleosome bound with FACT will not (Supplementary Fig. 5c). The different information of extensions of nucleome at 6pN can distinguish whether FACT binds to ubH2B-nucleosome. The calculated $K_d$ is $16.3 \pm 3.9$ nM (mean±SE, 1000 stretching cycles for 30 samples) for the binding of FACT with unmodified nucleosome, and $8.0 \pm 4.2$ nM (mean ± SE, 980 stretching cycles for 30 samples) for the binding of FACT with ubH2B nucleosome with significant difference $p < 0.01$ (Supplementary Fig. 5d).

### Real-time depostion assay of H2A-H2B dimer or H2A-ubH2B dimer on (H3-H4)$_2$ tetrasome by single-molecule magnetic tweezers

To confirm the chaperone property of FACT on H2B- or ubH2B-nucleosome, we traced the real-time deposition of H2A-H2B or H2A-ubH2B dimer on tetrasome in the presence of FACT, respectively. We anchored the mono-(H3/H4)$_2$ tetrasome with magnetic tweezers, and injected 100 μl FACT with H2A/H2B or H2A-ubH2B dimer (83 nM) into the flow cell, and traced the deposition process immediately at 1.5pN. Two sequential 10-nm deposition processes occurred, corresponding to the formation of the two halves of the outer DNA wrap, respectively. We then carried out the force-extension measurement for the same reassembled sample.

### mESC culture

The mouse embryonic stem cell (mESC), R1 cell line (ATCC® SCRC1011™) was obtained from American Type Culture Collection (ATCC) and used for biochemical, cellular and genomic analyses. R1 cells were cultured in the medium with 80% DMEM, 15% FBS, 1% Lglutamine, 1% nucleosides, 1% nonessential amino acids, 1% 2-mercaptoethanol, 1% Pen/Strep, and 1000U/ml leukemia inhibitory factor (LIF) in standard incubator with 5% CO$_2$ at 37 °C.

### ChIP analysis

mESCs were treated with tRA for 15 min, 30 min, 1 hr, 2 h, 4 h, 8 h, and then fixed with 1% formaldehyde. The fixed mESCs were resuspended in nuclei lysis buffer (50 mM Tris-HCl, pH 8.0, 10 mM EDTA, 1% SDS, protease inhibitors) and sonicated to 300–500-bp-sized fragments. 10 μg of the sonicated chromatin sample were used as input to calculated the relative enrichment of histone modifications or proteins on genome. Protein A (Life Technology, 10002D, 10 μl) and protein G Dynabeads (Life Technology, 10003D, 10 μl) were pre-incubated with H2B antibody (Abcam, ab1790, 2 μg), ubH2B antibody (CST, #5546,

2μg) and SSRP1 antibody (BioLegend, 609710, 2 μg) for 6 h at 4 °C and then incubated with 100 μg of the sonicated chromatin sample overnight at 4 °C. The beads were then extensively washed with RIPA150 (50 mM Tris-HCl, pH 8.0, 150 mM NaCl, 1 mM EDTA, pH 8.0, 0.1%SDS, 1%Triton X-100, 0.1% sodium deoxycholate) once, RIPA500 (50 mM Tris-HCl, pH 8, 0.5 M NaCl, 1 mM EDTA, pH 8.0, 0.1% SDS, 1%Triton X-100, 0.1%sodium deoxycholate) twice, RIPA-LiCl (50 mM Tris-HCl, pH 8.0, 1 mM EDTA pH 8.0, 1% NP-40, 0.7% sodium deoxycholate, 0.5 M LiCl) twice and 1×TE twice, and resuspended in freshly made elution buffer (10 mM Tris-HCl, pH 8.0, 300 mM NaCl, 5 mM EDTA, pH 8.0, 0.5% SDS). The mix was vortexed at 65 °C for 30 min and placed on a magnetic stand for 2 min. The supernatant was collected and de-crosslinked by incubation in 1% SDS at 65 °C for 6 h, followed by treatment with RNase A and proteinase K. The ChIPed and input DNA was extracted with phenol: chloroform: isoamyl (25:24:1) and precipitated with 75% ethanol and 300 mM NaAc followed by qPCR analysis. All the qPCR experiments were performed on the real-time PCR (ABI 7300, USA) using FastStart Universal SYBR Green Master (ROX). The primer pairs used for the qPCR experiments were the following: PHRF1: sense 5'-GGATGGGGAGTCGGACTGTA- 3', antisense 5'-AGTGC ACCCATACTCTTCCG- 3'; SIPA1: sense 5'-TCCACTACGTACCGCCTC-CATCCTGG- 3', antisense 5'-AGTACCTCTCGGGCTCAAGC- 3'; ERF: sense 5'-GATATTAACCCGGGAGGCGG- 3', antisense 5'-CTGTGTCCGC CGGGGTCTTC- 3'; SXRN1: sense 5'-CCAAGGAAGAGGTATGGGGC- 3', antisense 5'-GGCGATTGGTACGTTGTGC- 3'. The ChIP enrichment were normalized to the input, caculated with the formula [ChIPed DNA/(input DNAx10)]x100%. The results were calculated by three independent replications.

### Isolation of mRNA and real-time PCR analysis

mESCs were harvested after treated with tRA for 15 min, 30 min, 1 hr, 2 h, 4 h, 8 h, and then the total RNA was extracted using TRIzol reagent (Invitrogen), and the first strand of cDNA was reverse-transcribed using 2 μg of RNA. cDNA products were used for quantitative real-time PCR using the SYBR Premix Ex Taq (Takara) to validate the efficiency of tRA induction. The primer sequences used for real-time PCR were as follows: ERF: sense 5'-CATGAAGACCCCGGCGGACACAG- 3', antisense 5'-CTTGATGACAAACTCCCCGTA-3'; PHRF1: sense 5'-GACAGAGGCGG GATCGTCTAG- 3', antisense 5'-ATCACTGGCCAGTTCTGAG- 3'; SIPA1: sense 5'-GCGGTGGCCCAGCTTGAGCCCG- 3', antisense 5'-TGCTCTGT GGGCCTGACTCTC- 3'; SRXN1: sense 5'-CTGCATCGCCACGGTGCA-CAACGTAC- 3', antisense 5'-TGCAGCTGCTGGTAGGCTGCAT- 3'. The results were calculated by three independent replications.

### Genome-wide data analysis

The SSRP1 ChIP-seq data, RNA-seq expression data, and ubH2B ChIP-seq data were downloaded from Gene Expression Omnibus (GEO: GSE90906, PRJNA604675, GSE153584 and PRJNA643279). All the ChIP-seq data were mapped to the mouse genome (mm9) using bowtie2[56] with the default parameters, and the input controls were used in the peak-calling ananlysis for the ChIP-seq data. The single-end RNA-seq reads were mapped to the Mus musculus mm9 gene annotation model using hisat2, with the default parameters, and converted to bam files in samtools. Fragments overlapping representative transcripts from annotated genes (genecode vM1) were counted in the featureCounts software. The normalization was performed with R script. Low quality reads and PCR replicates were removed by samtools[57] and only uniquely mapped reads which mapping to a unique genomic location and strand were kept. Enriched peaks were called using MACS[58]. The p-value cutoff for peak detection was 1e-5 with MACS2.

### Reporting summary

Further information on research design is available in the Nature Portfolio Reporting Summary linked to this article.

## Data availability

The data that support this study are available from the corresponding authors upon reasonable request. Publicly available sequencing data analyzed in this study were downloaded from Gene Expression Omnibus under the accession ID number GSE90906 for the SSRP1 ChIP-seq data, GSE153584 for the ubH2B ChIP-seq data, PRJNA604675 and PRJNA643279 for the RNA-seq expression data. Source data are provided with this paper.

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

## Acknowledgements

This work was supported by grants from the National Natural Science Foundation of China (32022014 and 31871290 to P.C.; 11874414 to W.L.; 31991161 and 31630041 to G.L.; 21991133 and 11874415 to P.-Y.W.), the Ministry of Science and Technology of China (2017YFA0504202 to G.L.; 2018YFE0203302 to P.C.), the Beijing Municipal Science and Technology Committee (Z201100005320013 to G.L.), and the Chinese Academy of Sciences (CAS) Key Research Program on Frontier Science (QYZDY-SSW-SMC020 to G.L.; QYZDB-SSW-SLH045 to W.L.), the CAS Strategic Priority Research Program of Chinese Academy of Sciences (XDB37000000 to W.L.). We are also indebted to the colleagues whose work could not be cited due to the limitation of space.

## Author contributions

P.C. and W.L. conceived and supervised the project. A.L. prepared the samples and performed the biochemical analysis. J.K. and X.X. performed the single-molecule magenetic tweezer and AFM analysis. J.C. performed the pull-down assay, ChIP and qPCR analysis. J.L. performed the genome-wide data analysis. X.L. purified the DUB-wt and DUB-mut proteins. C.L., G.L. and P.-Y.W. helped to discuss the project. A.L., P.C. and W.L. analyzed the data and wrote the manuscript with input from all co-authors.

## Competing interests

The authors declare no competing interests.
