## [Peer Review File · Nature Communications]

REVIEWER COMMENTS

Reviewer #1 (Remarks to the Author):

The manuscript by Luo et al describes the study of the collaborative effect of H2B histone mono-ubiquitylation at lysine 120 and histone chaperone FACT on the mechanical stability of nucleosomes and gene activation process using nucleosome disruption experiments with magnetic tweezers, time-course ChIP-qPCR and analysis of ChIP-seq data. Using an elegant experimental system, the authors show that ubH2B impairs the mechanical stability of nucleosome and helps to recruit FACT by enhancing the binding of FACT on nucleosome. Binding of recruited FACT enhances the nucleosome stability and integrity and likely plays a role in gene activation.

This study addresses an important aspects of the mechanism of gene regulation, the data are of high quality and suggest an important mechanism likely utilized during chromatin transcription. However, some previously published data question biological significance of the proposed mechanism. This and other comments are described in detail below:

- 1) The role of H2B ubiquitination in position 120 on nucleosome disruption process was previously studied by Chen et al (doi: 10.7554/eLife.48281). The disruption of the ubH2B-containing nucleosomes was evaluated using dual-trap optical tweezers single-molecule unzipping assay that is considered as a more physiologically relevant assay than the pulling assay used in the current work. ubH2B nucleosomes showed quite similar disruption dynamic to wild type ones, while in the study of Luo et al the ubiquitination of H2B demonstrates the dramatic changes in the nucleosome stability. First, this previous work should be cited in the manuscript. Second, what is a possible reason for the apparent controversy? Finally, what is the biological significance of the observations on the different stability of the nucleosomes described in the manuscript? It should be further supported using an alternative, more physiologically relevant assay like a transcription assay.
- 2) Fig. 3: Interpreting the results of experiments about deposition of H2AH2B dimers (figure 3), the authors mention about the recovery of the complete nucleosomal structure. What is the evidence that it is a complete nucleosome and not a subnucleosome (for example, a hexasome)?
- 3) Figure 4 and S3: How was the level of total H2B measured? In Methods section only antibodies for uH2B and SSPR1 are described.
- 4) Figure 4 and ChIP-seq analysis: The original datasets used in the study include "input" controls. Did the authors used "input" controls during calculation of enrichment and peak-calling?
- 5) Figure 4B: Figure legend does not describe that read density is calculated for SSPR1. It is described in text and it will be very helpful to add the information in the figure legend.
- 6) Figure 4C: It will be very important to add the plots for ubH2B-only and ubH2B-desert.
- 7) One of the key intermediates in the model provided on the Figure 5 and in author's previous work on ubH2A role on FACT action (doi: 10.1093/nar/gkab1271) is a tetrasome. Is there any information about the tetrasome structures being formed at promoter regions? Can the processes described in the model occur in the absence of tetrasome structures?
- 8) For some figures increasing concentrations of components designated by the black triangles cannot be found in Methods or Results section or elsewhere (e.g. for figure 3B).

Reviewer #2 (Remarks to the Author):

The manuscript by Luo and colleagues investigates FACT engagement and chaperoning of nucleosomes with H2B ubiquitination. They use a magnetic tweezing approach they previously established (Chen et al. Mol Cell 2018, ref 25) to study the energetics of

nucleosome unwrapping when applying increasing force under different experimental conditions. The authors previously demonstrated that FACT lowers the nucleosome unwrapping energy barriers, leads to additional wrapping sub-steps, and supports at least three sequential wrapping cycles without loss of histones.

In their new study, the authors extend this powerful approach to nucleosomes with H2B ubiquitination and combine their single-molecule work with pulldowns, deposition assays, AFM, and ChIP-qPCR experiments. They observe that ubiquitination of H2B lowers the force required for each unwrapping transition (fig. 1D) in the absence of FACT. Very surprisingly, they observe a stabilization in the presence of FACT with higher forces required for unwrapping of ubH2B nucleosomes (fig. 2B). This is the opposite influence when compared to their previous report where FACT destabilized unmodified nucleosomes. Finally, they show FACT binds more strongly to ubH2B, prefers to deposit ubH2B, and ubH2B helps to recruit FACT for transcription activation based on ChIP-qPCR experiments.

Taken together, the authors provide many new insights into modulation of the engagement and chaperoning activities of FACT by histone modifications and ubH2B, specifically. Based on their magnetic tweezing experiments, the authors argue FACT alters the nucleosome state to help with gene activation (fig 5). However, the relative importance of ubH2B for recruitment versus altered nucleosome structure remains unclear to this reviewer. The work would be greatly strengthened if the authors could distinguish between these two important influences on FACT activity. What is the relative importance of recruitment due to ubH2B recognition versus differences in FACT engagement due to the structural changes resulting from the ubiquitin modification? Please see below for further specific comments.

Specific Comments

1. As stated in my general comments, it remains unclear whether the primary influence on FACT activity results from nucleosome structural changes resulting from ubiquitination or as a consequence of increased FACT recruitment due to ubH2B recognition. The manuscript would be greatly strengthened if the authors would clarify their arguments with regards to this point throughout the text and provide additional support for one or the other possibility. Definitive clarification could come from disrupting FACT recognition of ubH2B using truncations or mutations. A truncation or mutation that disrupts the recognition observed in the pulldowns, but retains the other activities could then be evaluated in the magnetic tweezing assays and potentially the ChIP-seq experiments to clarify the importance of these two pathways.
2. The authors commented that a very different activity is observed with ubH2A. Why is this the case? Could this offer clues about the importance of recruitment vs structural changes in the nucleosome due to ubiquitination? Presumably, ubH2A would enhance recruitment of FACT to the same extent as ubH2B, but with different alterations to nucleosome stability. Could comparative studies with ubH2A address my comment 1?
3. Do the authors observe substeps in the unwrapping transition in the presence of FACT similar to those observed in their earlier report (Chen et al. Mol Cell 2018, ref 25)? I could not see these substeps/half steps. However, they might be obscured due to the use of 3D plots. The authors should comment on the presence of unwrapping substeps. It would be informative to know how ubH2B influences these.
4. Displaying the repeated unwrapping cycles using 3D plots in figures 1E, 2B, 2E, 3C, 3D, 3F, 3G makes it very hard to compare differences in force between each cycle. The authors should use 2D plots that allow for direct comparison of small energy differences, which is a huge strength of their approach. If there is a strong justification for using 3D plots, the authors should add 2D versions of all plots to the supplement.
5. The H2A band is not visible in the fourth lane from the end in Figure 3A. This is a critical lane for comparison of relative differences between ubH2B and H2B, the authors should repeat this experiment and clarify why H2A is missing in this lane.

6. Could you clarify why there are input differences for Spt16 in Figure 2A?
7. Could you clarify why no band is visible in lane 7 (tetrasome and ubH2B)?
8. Line 244: "irreversible two-step unfolding dynamics" – The authors should comment on the single unfolding event in the second extension round (see Fig 1E) and how this relates to the histones staying in close proximity vs. being lost irreversibly. Is this event to be due to tetrasome unfolding?

Minor Comments

9. Fig 2B and E: Could you comment on the different rupture behaviour of the two populations? As the sample in E) is deubiquitinated, one would expect very similar traces?
10. Fig 2E, F, 3F, G: Please unify the axis increments between the figures
11. Fig 2E, F as well as 3D, E, 3G, H: Could you display traces more representative of the statistical analysis or comment on the differences?
12. Fig 2F: Could you add the force of H2B + FACT in absence of any DUB for direct comparison?
13. Fig 3F/G – Why is the sequence of conformational states inverted in the cartoons below 3F and 3G? Was that intended? Reversing 3G would be more accurate, since the nucleosome should be wrapped at low force. Unless the authors intended a different meaning.
14. Fig 4C: What does "desert" stand for?
15. Fig 4D: What do the numbers (e.g., 0-2.00) signify?
16. Fig 4E/F: Do the values signify an increase of the input in % on top of the starting concentration of 100 %?
17. Fig 5, 6: Could you change the colour of ubiquitin or the tetrasome to avoid confusion?
18. Fig S4: Would it be possible to show the absolute extension values?
19. Line 122: "please find more calculation details in the SI" – is it possible you refer to the methods section?
20. Line 137: "FACT greatly impairs ... but helps to maintain..."
21. Line 181: Would it maybe be more exact to use the median values instead of the higher borders of the upper quartile? In this case, e.g., about 16 pN instead of 20 pN?
22. Line 191: Are you referring to Figures 2B/C?
23. Line 193: "Once the intact nucleosome is formed..." – Is it possible that this somehow contradicts your theory of FACT interacting with ubH2B and depositing it? Are you able to show this sequence of events?
24. Line 213: "specific and typical gene regions" – could you quickly elaborate why you chose these gene regions?
25. Line 217/218: "genome-wide analysis showed that ubH2B helps to recruit more FACT to facilitate gene transcription" – What do you compare it to?
26. Line 227/228: Could you elaborate on how the H2B level stays relatively constant, while a new population of ubH2B arises?
27. Line 282-294: This could be very nice in the introduction
28. Line 593 ff: Would you unify the tenses used in the methods section?
29. Line 647: "equilibrium force is read directly $F_{eq}=3.5$ pN from Figure S2 right panel" – S4 right?

Reviewer #3 (Remarks to the Author):

In Luo et al, the authors investigate a long standing question of how ubH2B impacts nucleosome structure, and how the FACT complex actually facilitates nucleosome stability in a ubH2B preferred manner as observed in the literature. They utilize a combination of biochemical reconstitution, molecular tweezer, and CHIP-seq experiments to show that FACT indeed preferentially engages a ubH2B nucleosome. Overall, the manuscript

is clear, and the experiments largely address the effects of FACT on chromatin stability, with some surprising observations. There are a few experimental questions that should be clarified or updated, but I believe this work should be published upon corrections.

As a minor point, there are a series of typographical and grammatical errors scattered throughout the manuscript that make it distracting to read. One outstanding example is the inconsistent use of "SSRP1" with "SSPR1".

Questions for authors

1. The authors show in figure 1D that FACT appears to impair the stability of the nucleosome, despite FACT having greater affinity for H2Bub nucleosomes in Fig 2A. This seems rather counterintuitive.

a. Could the authors demonstrate using a more quantitative method to show affinities of FACT for nucleosomes? A gel shift assay like the one seen in Fig 3B, or Biolayer interferometry could provide some numbers that could provide additional confidence to this surprising finding.

2. The authors show quite convincingly that FACT appears to have a greater affinity for H2A-uH2B dimers compared to unmodified. However, rather surprisingly their gel shift assay appears to indicate that uH2B has no impact on the FACT dependent nucleosome assembly, at odds with Fig 3H.

a. It is a bit difficult to compare the results from their molecular tweezer experiments with their gel-based assays given the dramatically different DNA lengths used (400 bp for tweezers, 100 for gel). One can imagine a significant DNA component for these complexes. It would be useful for the authors to utilize a consistent DNA length across their experiments so that equivalent conclusions can be drawn from the data.

3. Can the authors clarify the comparison between Fig 2B and 2E, why there appears to be a significant shift in the inner wrap force on uH2B treated with DUB? It looks in 2B that the force for the inner wrap is under 10 pN, while in 2E the inner wrap is disrupted at forces > 10 pN despite it supposedly being an "identical" sample.

4. Similarly, to point 3, there is a very significant shift of the outer wrap force in the DUB mut FACT in 2F compared to the uH2B-FACT experiments in 2C. The quantification does not appear to reflect this. Can the authors comment on this difference.

Minor points

1. Please go through the manuscript carefully to correct typographical errors (chaperon vs chaperone, SSRP1 vs SSPR1, etc).

2. For the rupture experiments in fig 1 and fig 2 please have consistent X-axis points. This would make for easier interpretation and comparison of data.

3. Typically nucleosome reconstitutions are shown with 5% native acrylamide gels to give sharper bands, not agarose. The resolution of the bands in Fig 1A are quite poor. Please re-run and show the reconstitution using an acrylamide gel.

4. The authors fail to cite multiple works showing the structural impact of uH2B on enzymes, including some highly significant findings. In addition to the Worden papers on Dot1L and COMPASS there are:

- Xue et al, Nature 2019
- Hsu et al, Mol Cell 2019
- Park et al, Nat Comm 2019
- Anderson et al, Cell Rep 2019
- Valencia-Sanchez et al, Mol Cell 2019
- Jang et al, Genes Dev 2019
- Yao et al, Cell Res 2019

Please cite accordingly.

Reviewer #1 (Remarks to the Author):

The manuscript by Luo et al describes the study of the collaborative effect of H2B histone mono-ubiquitylation at lysine 120 and histone chaperone FACT on the mechanical stability of nucleosomes and gene activation process using nucleosome disruption experiments with magnetic tweezers, time-course ChIP-qPCR and analysis of ChIP-seq data. Using an elegant experimental system, the authors show that ubH2B impairs the mechanical stability of nucleosome and helps to recruit FACT by enhancing the binding of FACT on nucleosome. Binding of recruited FACT enhances the nucleosome stability and integrity and likely plays a role in gene activation.

This study addresses an important aspect of the mechanism of gene regulation, the data are of high quality and suggest an important mechanism likely utilized during chromatin transcription. However, some previously published data question biological significance of the proposed mechanism. This and other comments are described in detail below:

Response: We thank the reviewer for all the positive comments and constructive suggestions. We have performed additional experiments to satisfy the reviewers' concerns and thoroughly revised our manuscript according to the reviewer's comments, which are described point-by-point as follows.

1) The role of H2B ubiquitination in position 120 on nucleosome disruption process was previously studied by Chen et al (doi: 10.7554/eLife.48281). The disruption of the ubH2B-containing nucleosomes was evaluated using dual-trap optical tweezers single-molecule unzipping assay that is considered as a more physiologically relevant assay than the pulling assay used in the current work. ubH2B nucleosomes showed quite similar disruption dynamic to wild type ones, while in the study of Luo et al the ubiquitination of H2B demonstrates the dramatic changes in the nucleosome stability. First, this previous work should be cited in the manuscript. Second, what is a possible reason for the apparent controversy? Finally, what is the biological significance of the observations on the different stability of the nucleosomes described in the manuscript? It should be further supported using an alternative, more physiologically relevant assay like a transcription assay.

Response: We thank the reviewer to point out this issue.

First, as the submission guideline for reference is up to 50, we are sorry that some previously published works are not included in the original manuscript. We have cited and discussed this work (Doi: 10.7554/eLife.48281) in the revised manuscript Page 3 and 10-11, as suggested by the reviewer.

Consistent with the work mentioned by the reviewer (Chen et al, Doi: 10.7554/eLife.48281), we found that ubH2B did not affect the unfolding dynamics of nucleosome under stretching tension and both the unmodified H2B- and ubH2B- nucleosome displayed a quite similar irreversible two-step unfolding pathway, as shown in Figure 1. At the same time, we found that ubH2B mildly weakened the mechanical stability of nucleosome, especially the interactions between the DNA and histone, which is consistent with previously published FRET (Fluorescence Resonance Energy Transfer)-based work (*J Am Chem Soc* 2012, 134, 19548-19551). To further understand the impact of ubH2B on transcription process, especially in the presence of FACT complex, which plays essential roles in gene transcription, we investigated the mechanical stability and unfolding dynamics of ubH2B nucleosome with FACT. Importantly, we found that ubH2B forms a novel stabilized nucleosome-like state with the help of FACT and facilitates gene transcription. Based on the single-molecule force spectroscopy experiments, the genome-wide study and time-course ChIP-qPCR, we confirmed that the stabilized ubH2B-containing

nucleosome-like state with FACT functions as a key structural platform for transcription process. Compared with the unmodified nucleosome, we revealed the similar stability and dynamics for ubH2B-nucleosome with the pulling assay, which is consistent with the work with the unzipping assay. In this work, we focus on the effect of FACT on ubH2B nucleosome. We revealed a dramatic stability change of unH2B nucleosome in the presence of FACT which formed a stable and key intermediate to facilitate gene transcription.

The entire mechanical property and folding dynamics of nucleosome can be identified directly by the pulling assay used in the current work, which helps us reveal the direct effect of FACT on ubH2B nucleosome. To further understand this process, we are now trying to resolve the high-resolution cryo-EM structure of ubH2B-nucleosome bound with FACT or Pol II, and also trying to set up an *in vitro* transcription assay and the relative single-molecular unzipping assay to clear this question. BUT we don't think we can finish these kinds of experiments within a short time frame. Definitely, it's a very important question, and we are going to address these important issues in our future works.

We have added the related comments with the indicated reference cited in the text, as shown in Page 3 and 10-11 in the revised manuscript.

2) Fig. 3: Interpreting the results of experiments about deposition of H2AH2B dimers (figure 3), the authors mention about the recovery of the complete nucleosomal structure. What is the evidence that it is a complete nucleosome and not a subnucleosome (for example, a hexasome)?

Response: This is a very good question. From the crystal structure of nucleosome, we know that the H3/H4 tetramer wrapped around 80 bp DNA, with one H2A/H2B dimer wrapped about more 30 bp at one side, and another H2A/H2B dimer wrapped more 30 bp at another side to form an intact nucleosome state. When an intact nucleosome is unfolded under tension, the outer DNA wrap bound to the two H2A/H2B dimers, about 2×10 nm step size will first release. So, to examine whether a nucleosome or a hexasome is formed by the deposition of H2A/H2B with the help of FACT, we can analyze the step sizes released by unfolding the formed structure after deposition.

As the reviewer concerned, to check whether a complete nucleosome is formed, we conduct the *in-situ* deposition experiment by magnetic tweezers. Here we use the deposition of H2A/ubH2B as an example. We tethered the mono-(H3/H4)₂ tetrasome with magnetic tweezers, and injected the H2A/ubH2B dimer with FACT into the flow cell, to observe the real-time deposition process. As shown in the panel a below, two sequential 10-nm deposition steps were observed, which corresponds to the sequential deposition of two H2A/ubH2B dimers. In addition, we carried out the force-extension measurement for the same molecule after the deposition. As shown in the panel b, an obviously two-step unfolding pathway of intact ubH2B-nucleosome with FACT was observed. These results further confirmed that FACT deposits H2A/ubH2B to form an intact nucleosome.

We have included the related issue in the revised manuscript Page 8, and add a new figure as shown in the revised Figure S3c-d.

3) Figure 4 and S3: How was the level of total H2B measured? In Methods section only antibodies for uH2B and SSPR1 are described.

Response: We thank the reviewer to point out this issue. We used the H2B antibody (ab1790) to detect the level of total H2B in this experiment. We have added the information in the revised Methods “ChIP analysis” section.

4) Figure 4 and ChIP-seq analysis: The original datasets used in the study include “input” controls. Did the authors used “input” controls during calculation of enrichment and peak-calling?

Response: Yes, we used the “input” controls in the peak-calling analysis for the ChIP-seq data of ubH2B and FACT. We have added the information in the revised Methods “Genome-wide data analysis” section.

5) Figure 4B: Figure legend does not describe that read density is calculated for SSRP1. It is described in text and it will be very helpful to add the information in the figure legend.

Response: We thank the reviewer to point out this issue. We have added the information in the revised Figure 4’s legend as suggested.

6) Figure 4C: It will be very important to add the plots for ubH2B-only and ubH2B-desert.

Response: We have added the plot and related information on Figure S4a. ubH2B and FACT cooperate with each other to activate gene transcription. As shown in the panel a below, for all the genes enriched with ubH2B, more ubH2B are enriched in the genes with SSRP1 overlapped, as compared to those with ubH2B-only. In addition, ubH2B & SSRP1 overlapped genes are significantly more active than the ubH2B only genes, and ubH2B containing genes are more active than ubH2B desert genes (panel b). We have added the information in Figure S4a and in the revised manuscript in Page 9.

7) One of the key intermediates in the model provided on the Figure 5 and in author’s previous work on ubH2A role on FACT action (Doi: 10.1093/nar/gkab1271) is a tetrasome. Is there any information about the tetrasome structures being formed at promoter regions? Can the processes described in the model occur in the absence of tetrasome structures?

Response: This is a very good question. The FACT complex is identified as a chaperone of histone H2A/H2B dimer and help to deposit H2A/H2B dimer to from nucleosome. As we revealed in Question 2, FACT can sequentially deposit H2A/H2B dimer onto tetrasome to form hexasome and then intact nucleosome. Actually, the *in vivo* nucleosome state is quite dynamic. The passage of RNA or DNA polymerase during transcription or replication will generate sub-nucleosome, including tetrasome and hexasome. Rhee et al (*Cell* 2014, 159, 1377-88) has revealed widespread

sub-nucleosomal structure in dynamic chromatin across the yeast genome by using ChIP-exo technique. Our previous study published in 2020 (*Nucleic Acids Research* 48, 5939-5952) also showed that the subnucleosome with H2A/H2B detached is highly regulated at promoter regions in mouse ES cells.

To examine the model in the absence of tetrasome, we test whether the process can occur on DNA template. Our work has shown FACT can bind DNA and H3/H4 histone, and can maintained the structure of preformed tetrasome (Figure 3c-d). But whether FACT can mediate the deposition of H3/H4 on DNA to form tetrasome is remained obscure. We try to examine the chaperone property of FACT on H3/H4 by using the supercoiling assay for *in vitro* assembly of tetrasome on DNA by FACT. As show in the figure below, 0.1 μg ϕX174 DNA (lane 1) was treated with DNA topoisomerase I (lane 2) and incubated with 0.1 μg of histone H3/H4 tetramer (lane 3, negative control), and with increasing amounts (0.25 and 0.5 μg) of NAP-1 (lanes 4,5, as positive control) or the same ratio of FACT (lanes 6,7). The results showed that as the NAP-1 can deposit H3/H4 on DNA to form tetrasome, FACT does not have this ability. So, we think FACT should function at the tetrasome level, that's why we use the tetrasome as a key intermediate in our model for understanding the FACT's function.

H3/H4	-	-	+	+	+	+	+
Topo I	-	+	+	+	+	+	+
DNA	+	+	+	+	+	+	+

8) For some figures increasing concentrations of components designated by the black triangles cannot be found in Methods or Results section or elsewhere (e.g., for figure 3B).

Response: We thank the reviewer to point out this issue. We have added all the related information in the revised methods as suggested.

Reviewer #2 (Remarks to the Author):

The manuscript by Luo and colleagues investigates FACT engagement and chaperoning of nucleosomes with H2B ubiquitination. They use a magnetic tweezing approach they previously established (Chen et al. Mol Cell 2018, ref 25) to study the energetics of nucleosome unwrapping when applying increasing force under different experimental conditions. The authors previously demonstrated that FACT lowers the nucleosome unwrapping energy barriers, leads to additional wrapping sub-steps, and supports at least three sequential wrapping cycles without loss of histones.

In their new study, the authors extend this powerful approach to nucleosomes with H2B ubiquitination and combine their single-molecule work with pulldowns, deposition assays, AFM, and ChIP-qPCR experiments. They observe that ubiquitination of H2B lowers the force required for each unwrapping transition (fig. 1D) in the absence of FACT. Very surprisingly, they observe a stabilization in the presence of FACT with higher forces required for unwrapping of ubH2B nucleosomes (fig. 2B). This is the opposite influence when compared to their previous report where FACT destabilized unmodified nucleosomes. Finally, they show FACT binds more strongly to ubH2B, prefers to deposit ubH2B, and ubH2B helps to recruit FACT for transcription activation based on ChIP-qPCR experiments.

Taken together, the authors provide many new insights into modulation of the engagement and chaperoning activities of FACT by histone modifications and ubH2B, specifically. Based on their magnetic tweezing experiments, the authors argue FACT alters the nucleosome state to help with gene activation (fig 5). However, the relative importance of ubH2B for recruitment versus altered nucleosome structure remains unclear to this reviewer. The work would be greatly strengthened if the authors could distinguish between these two important influences on FACT activity. What is the relative importance of recruitment due to ubH2B recognition versus differences in FACT engagement due to the structural changes resulting from the ubiquitin modification? Please see below for further specific comments.

Response: We thank the reviewer for all the constructive critiques, helpful suggestions and comments. We have carefully examined the reviewer's comments and try to clarify and/or perform additional experiments to satisfy the reviewers' concerns in the revised manuscript. We describe the revision point-by-point as follows.

Specific Comments

1. As stated in my general comments, it remains unclear whether the primary influence on FACT activity results from nucleosome structural changes resulting from ubiquitination or as a consequence of increased FACT recruitment due to ubH2B recognition. The manuscript would be greatly strengthened if the authors would clarify their arguments with regards to this point throughout the text and provide additional support for one or the other possibility. Definitive clarification could come from disrupting FACT recognition of ubH2B using truncations or mutations. A truncation or mutation that disrupts the recognition observed in the pulldowns, but retains the other activities could then be evaluated in the magnetic tweezing assays and potentially the ChIP-seq experiments to clarify the importance of these two pathways.

Response: We thank the reviewer to point out this issue. It's a very interesting question. What we can conclude now is that this effect is not caused by the modified ubiquitin itself. Our previous investigations on the effect of ubH2A on nucleosome and FACT's function has shown that the ubiquitination of H2A greatly stabilized the nucleosome state, and interestingly, ubH2A blocks the binding of FACT on nucleosome (*J Am Chem Soc* 2020, 142, 3340-3345; *Nucleic Acids Research* 2022, 50, 833-846). As we discussed in last two paragraph of this manuscript, the same "ubiquitin" modified on nucleosome has totally different effect on FACT's function. We believe the specific sites where "ubiquitin" modified at nucleosome plays essential roles here. Recent cryo-EM structure of the sub-nucleosome-FACT complex revealed by Liu et al (*Nature* 2020, 577, 426-431)

showed that the dimerization domains of SPT16 and SSRP1 straddle on the dyad of nucleosome with the middle domains of SPT16 and SSRP1 projecting down on the either side of nucleosome's face, as like a unicycle. The ubiquitin modified at H2A-K119 site is located at the beginning of the C-terminal tails of H2A at DNA entry/exit point around the dyad of nucleosome, which blocks the peeling of DNA from the octamer to stabilize the nucleosome, and also clash and exclude the binding of FACT on nucleosome, as we revealed in previous studies (*J Am Chem Soc* 2020, 142, 3340-3345; *Nucleic Acids Research* 2022, 50, 833-846). But for ubH2B, the ubiquitin modified at H2B-K120 sits on the face of nucleosome, which may contact with the middle domains of SPT16 or SSRP1 to stabilize the binding of FACT on nucleosome and affect its function.

As suggested by the reviewer, we are trying to clarify which part of FACT functions for the specific recognition of ubH2B. We conduct the mono-nucleosome pulldown assay using SPT16 or SSRP1, the subunit of FACT. As shown in the figure below, our results showed that ubH2B has little effect on the binding of SPT16 with nucleosome (panel a), but does enhance the binding of SSRP1 with nucleosome (panel b). Actually, to further clarify this process in more details, we are also trying to resolve the high-resolution cryo-EM structure of ubH2B-nucleosome bound with FACT or SSRP1, and trying to understand how ubH2B coordinates FACT to fulfill this function. BUT we don't think we can finish these experiments within a short time frame. Definitely, we are going to address these important issues in our future works. We have discussed this issue in Page 12 in the revised manuscript.

2. The authors commented that a very different activity is observed with ubH2A. Why is this the case? Could this offer clues about the importance of recruitment vs structural changes in the nucleosome due to ubiquitination? Presumably, ubH2A would enhance recruitment of FACT to the same extent as ubH2B, but with different alterations to nucleosome stability. Could comparative studies with ubH2A address my comment 1?

Response: This is a very good question. As we discussed at the above question 1, the same “ubiquitin” modified on nucleosome does have totally different effect on FACT’s function. Our previous investigations on the effect of ubH2A on nucleosome and FACT’s function has shown that the ubiquitination of H2A greatly stabilized the nucleosome state and interestingly, ubH2A blocks the binding of FACT on nucleosome (*J Am Chem Soc* 2020, 142, 3340-3345; *Nucleic Acids Research* 2022, 50, 833-846), totally different from that of ubH2B. We believe the specific sites where “ubiquitin” modified at nucleosome plays essential roles here. According to the recent cryo-EM structure of the sub-nucleosome-FACT complex revealed by Liu et al (*Nature* 2020, 577, 426-431), the dimerization domains of SPT16 and SSRP1 straddle on the dyad of nucleosome with the middle domains of SPT16 and SSRP1 projecting down on the either side of nucleosome’s face, as like a unicycle. The ubiquitin modified at H2A-K119 site is located at the beginning of the C-terminal tails of H2A at DNA entry/exit point around the dyad of nucleosome, which blocks the

peeling of DNA from the octamer to stabilize the nucleosome, and also clash and exclude the binding of FACT on nucleosome, as we revealed in previous studies (*J Am Chem Soc* 2020, 142, 3340-3345; *Nucleic Acids Research* 2022, 50, 833-846). But for ubH2B, the “ubiquitin” modified at H2B-K120 is sitting on the face of nucleosome, which may contact with the middle domains of SPT16 or SSRP1 to stabilize the binding of FACT on nucleosome and affect its function. The high-resolution cryo-EM structure of ubH2B-nucleosome bound with FACT can clarify this effect in more details. We are now trying to resolve this structure, BUT we don't think we can finish these experiments within a short time frame. Definitely, we are going to address these important issues in our future works.

3. Do the authors observe substeps in the unwrapping transition in the presence of FACT similar to those observed in their earlier report (Chen et al. *Mol Cell* 2018, ref 25)? I could not see these substeps/half steps. However, they might be obscured due to the use of 3D plots. The authors should comment on the presence of unwrapping substeps. It would be informative to know how ubH2B influences these.

Response: We thank the reviewer to point out this issue. Actually, the ubH2B does not influence the unfolding substeps in the presence of FACT. As shown in the figure below, similar substeps can be observed for the unfolding process of ubH2B-nucleosome as that for unmodified nucleosome in the presence of FACT. The substeps during disruption indicate the asymmetric mechanical property of the DNA wrapped around histone octamer. Due to the short life time and the sampling limitation of CCD, we cannot always record these substeps for every nucleosome sample. In this work, we focused on the effect of FACT on the mechanical stability of ubH2B-nucleosome. The substeps in the disruption of the outer and inner DNA wrap have little effect on the determination of mechanical stability of nucleosome. We agree with the reviewer and we are trying to update our magnetic tweezers to reveal the folding and unfolding dynamics of nucleosome with higher temporal resolution.

4. Displaying the repeated unwrapping cycles using 3D plots in figures 1E, 2B, 2E, 3C, 3D, 3F, 3G makes it very hard to compare differences in force between each cycle. The authors should use 2D plots that allow for direct comparison of small energy differences, which is a huge strength of their approach. If there is a strong justification for using 3D plots, the authors should add 2D versions of all plots to the supplement.

Response: We thank the reviewer for the good suggestion. We have added all the 2D versions of the Figure 1e, 2b, 2e, 3c, 3d, 3f, 3g in the related supplementary Figure S1-3 in SI as suggested.

5. The H2A band is not visible in the fourth lane from the end in Figure 3A. This is a critical lane for comparison of relative differences between ubH2B and H2B, the authors should repeat this experiment and clarify why H2A is missing in this lane.

Response: We thank the reviewer to point out the issue. There is histone H2A in the band, but looks a little bit weak in the original Figure 3a (panel a as follows). In the panel b, we show the results for the same sample with longer exposure time, which can clearly see the H2A band. To avoid confusion, we have replaced the lane with longer exposure time as shown in revised Figure 3a.

6. Could you clarify why there are input differences for Spt16 in Figure 2A?

Response: That's caused by the edge effect, frequently observed in western blot experiments. As shown in the original experimental film as following figure panel a, the input of Spt16 (lane 1) is at the edge of the gel. Although we input the same quantity of Spt16 as that in lane 3 and 5, one edge of the gel will frequently look less than the others. But it does not affect our results. We have replaced the lane by that from the same sample with longer exposure time, which looks better as shown in the following panel b in revised Figure 2a.

7. Could you clarify why no band is visible in lane 7 (tetrasome and ubH2B)?

Response: Without the help of histone chaperone FACT, H2A/ubH2B dimer cannot be deposited into tetrasome to form a homogeneous complex shown as a band in gel. The H2A/ubH2B dimer will randomly interact with tetrasome, and form aggregations. That's why in lane 7, no band is visible, but we can see the aggregated samples in PAGE gel hole at top.

8. Line 244: “irreversible two-step unfolding dynamics” – The authors should comment on the single unfolding event in the second extension round (see Fig 1E) and how this relates to the histones staying in close proximity vs. being lost irreversibly. Is this event to be due to tetrasome unfolding?

Response: We thank the reviewer to point out the issue. The previous studies of ours and other groups have revealed that the intact nucleosome unfolds in a two-step process under tension, which corresponds to the unraveling of the outer nucleosomal DNA wrap, mainly stabilized by the interactions between H2A/H2B dimer and DNA, and the unraveling of the inner DNA wrap, mainly stabilized by the interactions between H3/H4 tetramer and DNA. To examine the reversibility of the structure, we fully unfolded the intact nucleosome under increasing tension up to 32 pN, and decreased the tension to 0.1 pN directly and waited for 5 minutes and then repeated the stretching cycle several times on the same nucleosome sample. For the unmodified or ubH2B nucleosome, the two-step unfolding pathway for intact nucleosome is only observed in the first

cycle. In the second round, we can observe some irregular one-step unfolding process, which may cause by some remaining histone bound on DNA. We do not know it's tetrasome or some other irregular state. but we know it's absolutely not an intact nucleosome. In the third round and after, the stretching modes are similar to that of free DNA, indicating that the histones are all displaced from the DNA after the repeated stretching process. We are sorry for the confusion. We have clarified this issue in the revised manuscript Page 5.

Minor Comments

9. Fig 2B and E: Could you comment on the different rupture behaviour of the two populations? As the sample in E) is deubiquitinated, one would expect very similar traces?

Response: We thank the reviewer to point out this issue. As we shown in the statistical analysis in Figure 2c and 2f, de-ubiquitination of ubH2B nucleosome do rescue the destabilized effect of FACT on nucleosome. The statistical analysis also revealed that the variation is a little bit larger in the DUB process, as we shown several behaviors of different molecules in the figure below; but the de-ubiquitination of ubH2B nucleosome with FACT do unwrap at much lower force than the ubH2B nucleosome with FACT. To avoid the misleading, we selected a more representative trace of the statistical analysis to display in the revised Figure 2.

10. Fig 2E, F, 3F, G: Please unify the axis increments between the figures

Response: We have unified all the axis in revised figures as suggested.

11. Fig 2E, F as well as 3D, E, 3G, H: Could you display traces more representative of the statistical analysis or comment on the differences?

Response: As we discussed in the question 9 pointed by the reviewer, we selected and displayed a more representative trace of the statistical analysis in the revised Figure 2 and Figure 3.

12. Fig 2F: Could you add the force of H2B + FACT in absence of any DUB for direct comparison?

Response: We thank the reviewer to point out this issue. We have the trace of the H2B+FACT in the absent of DUB in Figure 2a. In addition, we add the trace of H2B-nucleosome + DUB (panel a) and H2B-nucleosome + DUB+ FACT (panel b) as the control as show in below, which indicated that the DUB has no effect on the nucleosome dynamics and the FACT's function on nucleosome. We showed the results in the revised Figure S2c-d and add the related information in Page 6 in

the revised manuscript.

13. Fig 3F/G – Why is the sequence of conformational states inverted in the cartoons below 3F and 3G? Was that intended? Reversing 3G would be more accurate, since the nucleosome should be wrapped at low force. Unless the authors intended a different meaning.

Response: We thank the reviewer to point out this issue. We are sorry for the misplacement of the conformational states. We have corrected the figure as shown in the revised Figure 3f/g.

14. Fig 4C: What does “desert” stand for?

Response: Here, “desert” means the gene region where has no SSRP1 or ubH2B ChIP signal peaks. We are sorry for the confusion and have added the information in the revised Figure 4 legend.

15. Fig 4D: What do the numbers (e.g., 0-2.00) signify?

Response: We are sorry for the confusion. The number (e.g 0-2.00) stands for the y-axis of the data, which is the read density ($\log_2(\text{RPM}+1)$) from the ChIP-seq data of SSRP1 and ubH2B, and gene expression ($\log_2(\text{RPM}+1)$) from RNA-seq data, respectively. RPM: the reads per million mapped reads. To make it clearer, we add the information in the revised Figure 4’s legend.

16. Fig 4E/F: Do the values signify an increase of the input in % on top of the starting concentration of 100 %?

Response: We thank the reviewer to point out this issue. We carefully examined the raw data. In each ChIP study, we took out 10 μg of chromatin sample before IP as input to calculate the relative enrichment of histone marks or proteins on genome, and 100 μg of chromatin sample used for ChIP experiment. Then the extracted ChIPed and input DNA were semi-quantified using real-time PCR equipment (ABI 7300). The relative enrichment values are calculated with the formula: $[\text{ChIPed DNA}/(\text{input DNA} \times 10)] \times 100 \%$. However, in right panel of original Figure 4 E-F, we forget multiplying each input DNA by 10 when processing the data, so the “% of input” values is 10 times larger than the truth value. We are very sorry for the errors and thank the reviewer to point out this issue. We have corrected them in the revised Figure 4 e-f, and added the related experimental details in the revised Methods “ChIP analysis” section.

17. Fig 5, 6: Could you change the colour of ubiquitin or the tetrasome to avoid confusion?

Response: We have changed the color of ubiquitin as suggested in the revised Figures.

18. Fig S4: Would it be possible to show the absolute extension values?

Response: We have showed the absolute extension values in the revised Figure S5a as suggested.

19. Line 122: “please find more calculation details in the SI” – is it possible you refer to the methods section?

Response: We are very sorry for the errors. We have corrected them in the revised manuscript.

20. Line 137: “FACT greatly impairs ... but helps to maintain...”

Response: We have modified the sentence to make it clearer.

21. Line 181: Would it maybe be more exact to use the median values instead of the higher borders of the upper quartile? In this case, e.g., about 16 pN instead of 20 pN?

Response: We thank the reviewer to point out this issue. We have revised this value as suggested.

22. Line 191: Are you referring to Figures 2B/C?

Response: We have revised the sentence to avoid confusion in the revised manuscript.

23. Line 193: “Once the intact nucleosome is formed...” – Is it possible that this somehow contradicts your theory of FACT interacting with ubH2B and depositing it? Are you able to show this sequence of events?

Response: We proposed that FACT prefers to interact with H2A/ubH2B dimer, and deposit the dimer into tetrasome to form nucleosome. After the deposition process, FACT remains to bind the ubH2b-nucleosome to maintain a stabilized nucleosome state. We have revised the sentence to avoid confusion in the revised manuscript.

24. Line 213: “specific and typical gene regions” – could you quickly elaborate why you chose these gene regions?

Response: We chose the typical gene region according to the genome-wide analysis in Figure 4a and c. First, our genome-wide analysis showed a total of 9786 SSRP1 enrichment sites by model-based analysis of ChIP-seq peak calling. Among them, 2489 SSRP1 peaks overlapped with ubH2B peaks. We then screened the gene region to choose the typical gene region, where enriched both SSRP1 and ubH2B; or only enriched with SSRP1, or no SSRP1 enriched, and then examined their related RNA expression level, and presented a typical one in Figure 4d.

25. Line 217/218: “genome-wide analysis showed that ubH2B helps to recruit more FACT to facilitate gene transcription” – What do you compare it to?

Response: In brief, first our genome-wide analysis showed a total of 9786 SSRP1 enrichment sites; among them, 2489 SSRP1 peaks overlapped with ubH2B peaks (Figure 4a). We can separate the SSRP1 regulated gene region to two groups: only SSRP1 without ubH2B, and both SSRP1 & ubH2B overlapped.

Second, we analyzed the read density of SSRP1 in these two groups, and found more SSRP1 are enriched in the gene regions containing ubH2B, as compared to those without ubH2B (Figure 4b). That indicated ubH2B helps to recruit more FACT.

Then we analyzed the expression level of these two groups. The genes containing both SSRP1 & ubH2B are more active than those only containing SSRP1 (Figure 4c). That suggested ubH2B

coordinate with SSRP1 to facilitate gene expression.

Overall, from the genome-wide analysis, we proposed that ubH2B helps to recruit more FACT to facilitate gene transcription. We have reorganized this part to avoid confusion in the revised manuscript.

26. Line 227/228: Could you elaborate on how the H2B level stays relatively constant, while a new population of ubH2B arises?

Response: Because the H2B antibody we used can interact with both unmodified H2B and ubH2B, while the ubH2B antibody only recognize the nucleosome with ubH2B. During the transcription process, the whole H2B level (recognized by the H2B antibody) does not changed, while the population of modified ubH2B increases. We have clarified this information in the revised figure legend to avoid the confusion.

27. Line 282-294: This could be very nice in the introduction

Response: We thank the reviewer to point out this issue. We have revised the manuscript as suggested in Page 3.

28. Line 593 ff: Would you unify the tenses used in the methods section?

Response: We thank the reviewer to point out this issue. We have thoroughly revised the methods section as suggested.

29. Line 647: “equilibrium force is read directly $F_{eq}=3.5$ pN from Figure S2 right panel” – S4 right?

Response: We are very sorry for the error. We have corrected it in the revised manuscript.

Reviewer #3 (Remarks to the Author):

In Luo et al, the authors investigate a long standing question of how ubH2B impacts nucleosome structure, and how the FACT complex actually facilitates nucleosome stability in a ubH2B preferred manner as observed in the literature. They utilize a combination of biochemical reconstitution, molecular tweezer, and CHIP-seq experiments to show that FACT indeed engages preferentially engages a ubH2B nucleosome. Overall, the manuscript is clear, and the experiments largely address the effects of FACT on chromatin stability, with some surprising observations. There are a few experimental questions that should be clarified or updated, but I believe this work should be published upon corrections.

As a minor point, there are a series of typographical and grammatical errors scattered throughout the manuscript that make it distracting to read. One outstanding example is the inconsistent use of “SSRP1” with “SSPR1”.

Response: We thank the reviewer for all the constructive suggestions and comments. We have carefully examined the reviewer’s comments and revised our manuscript accordingly with additional experiments to address the reviewer’s concerns as follows.

Questions for authors

1. The authors show in figure 1D that FACT appears to impair the stability of the nucleosome, despite FACT having greater affinity for H2Bub nucleosomes in Fig 2A. This seems rather counterintuitive.
a. Could the authors demonstrate using a more quantitative method to show affinities of FACT for nucleosomes? A gel shift assay like the one seen in Fig 3B, or Biolayer interferometry could provide some numbers that could provide additional confidence to this surprising finding.

Response: In Figure 1d, we showed that compared to the unmodified nucleosome, ubH2B mildly impairs the stability of the nucleosome. In Figure 2a, we found more FACT binds to ubH2B nucleosome, as compared to the unmodified nucleosome. We are trying to give a more quantitative method to show the binding affinity of FACT on nucleosome as suggested.

This is an excellent suggestion. We do try to find a more quantitative method to show the binding affinity of FACT on nucleosome. We tried to use the MicroScale Thermophoresis (MST) methods to measure the binding capacity of FACT on nucleosome as shown in below figure. We fluorescently labeled the H2B and ubH2B-nucleosome (panel a), and can use the MST to measure the binding of DUB with nucleosome. BUT when we tried to measure the K_d of FACT binding with the nucleosomes, we failed to get the valid data unfortunately (panel c).

We then try to measure the K_d for FACT binding to unmodified H2B or ubH2B-nucleosome by using the magnetic tweezers, according to the method published in our previously work (Molecular Cell 2018, 71, 284-293). In brief, the mechanical stability of nucleosome will be changed when FACT binds to the H2B or ubH2B- nucleosome. Here we use the binding of H2B-nucleosome with FACT as an example. Based on the results, we exerted a proper tension ($F = 11$ pN) where the H2B-nucleosome alone is partial disrupted, but the nucleosome bound with FACT is totally disrupted, to distinguish the two fractions: the nucleosome alone and the nucleosome bound with FACT. The extension we read out is the index for different state of the nucleosome. We performed the force-jump measurements for the samples in the flow cell as many as possible and counted the number for the nucleosome alone ($N_{nucleosome}$) and the complex of nucleosome bound with FACT or its component ($N_{complex}$). Based on the definition of dissociation constants, we can obtain K_d value of FACT through the equation:

$$k_d = \frac{[nucleosome][FACT]}{[complex]} = \frac{[nucleosome]}{[complex]} [FACT] = \frac{N_{nucleosome}}{N_{complex}} [FACT]$$

By using this method, as shown in figure below, we calculated the K_d of FACT binding with the H2B-nucleosome is 16.4 ± 5 nM (mean \pm SE, N=339), and the K_d of FACT binding with the ubH2B-nucleosome is 10.2 ± 4 nM (mean \pm SE, N=110). The results are consistent with our mono-nucleosome pull-down assay that ubH2B promotes the binding of FACT with nucleosome. We have added the details in Page 6 in the revised manuscript, the Method “The dissociation constant K_d measured by single molecule magnetic tweezers” section and Figure S5b-c.

2. The authors show quite convincingly that FACT appears to have a greater affinity for H2A-uH2B dimers compared to unmodified. However, rather surprisingly their gel shift assay appears to indicate that uH2B has no impact on the FACT dependent nucleosome assembly, at odds with Fig 3H.

a. It is a bit difficult to compare the results from their molecular tweezer experiments with their gel-based assays given the dramatically different DNA lengths used (400 bp for tweezers, 100 for gel). One can imagine a significant DNA component for these complexes. It would be useful for the authors to

utilize a consistent DNA length across their experiments so that equivalent conclusions can be drawn from the data.

Response: We thank the reviewer to point out this issue. In the Figure 3a, we showed that FACT prefers to bind H2A-ubH2B compared to the unmodified dimer. We then examined the effect of ubH2B on the binding of FACT-H2A-H2B on (H3-H4)₂ tetrasome, as shown in Figure 3b. Actually, the ubH2B looks very slightly enhance the binding property of FACT-H2A-H2B on (H3-H4)₂ tetrasome, but does not have a pronounced effect. We can see with the addition of FACT-H2A-H2B, the band of tetrasome shifted to the band related to the complex formed by FACT-H2A-H2B with (H3-H4)₂ tetrasome. If we observed the missing rate of the tetrasome band for ubH2B is a little faster than that for unmodified H2B. In Figure 3h, we investigated whether the H2A/H2B or H2A/ubH2B can be deposit for form a nucleosome by FACT. We observed the different property of FACT by different methods, Figure 3a and 3b to observe the binding property, Figure 3h the chaperone property.

For the different DNA lengths problem, different method needs different DNA to optimize the experimental results. For example, for gel analysis, short DNA length less than 200 bp helps to reconstitute homogeneous mono-nucleosome samples, and minimize the nonspecific interaction of proteins with bared DNA outside the nucleosome. But in single-molecule magnetic tweezer experiments, the nucleosome DNA needs to attach on a glass coverslip at one end, and a 2.8- μ m diameter superpara-magnetic Dynabead at the other end. A longer DNA length of at least 400 bp is needed to avoid the nonspecific interaction of the bead with the cover glass's surface.

3. Can the authors clarify the comparison between Fig 2B and 2E, why there appears to be a significant shift in the inner wrap force on uH2B treated with DUB? It looks in 2B that the force for the inner wrap is under 10 pN, while in 2E the inner wrap is disrupted at forces > 10 pN despite it supposedly being an “identical” sample.

Response: We thank the reviewer to point out this issue. As we shown in the statistical analysis in Figure 2c and 2f, de-ubiquitination of ubH2B nucleosome do rescue the destabilized effect of FACT on nucleosome. The statistical analysis also revealed that the variation is a little bit larger in the DUB process, as we shown several behaviors of different molecules in the figure below. There are some molecules disrupted the inner wrap at force > 10 pN, but the de-ubiquitination of ubH2B nucleosome with FACT do unwrap at much lower force than the ubH2B nucleosome with FACT. To avoid the misleading, we selected a more representative trace of the statistical analysis to display in the revised Figure 2.

4. Similarly, to point 3, there is a very significant shift of the outer wrap force in the DUB mut FACT in 2F compared to the ubH2B-FACT experiments in 2C. The quantification does not appear to reflect this. Can the authors comment on this difference.

Response: As we discussed in point 3, if we compare the statistical analysis in Figure 2c and 2f, the adding of DUB-wt on ubH2B-nucleosome with FACT is similar to that of H2B nucleosome with FACT, and the adding of DUB-mut is similar to ubH2B-nucleosome, which indicates the de-ubiquitination of ubH2B will rescue the FACT's function on nucleosome. But as the reviewer pointed out, the variation is a little bit larger in the DUB-wt or DUB-mut process. To avoid the misleading, we selected a more representative trace of the statistical analysis to display in the revised Figure 2.

Minor points

1. Please go through the manuscript carefully to correct typographical errors (chaperon vs chaperone, SSRP1 vs SSPR1, etc).

Response: We are very sorry for the grammatical and stylistic errors. We have checked our manuscript sentence by sentence and carefully modified our manuscript.

2. For the rupture experiments in fig 1 and fig 2 please have consistent X-axis points. This would make for easier interpretation and comparison of data.

Response: We thank the reviewer to point out this issue. We have unified all the axis in revised figures as suggested.

3. Typically nucleosome reconstitutions are shown with 5% native acrylamide gels to give sharper bands, not agarose. The resolution of the bands in Fig 1A are quite poor. Please re-run and show the reconstitution using an acrylamide gel.

Response: We have re-run the gels with 5% native acrylamide gels as suggested in figure below. The data are shown in revised Figure 1a.

4. The authors fail to cite multiple works showing the structural impact of uH2B on enzymes, including some highly significant findings. In addition to the Worden papers on Dot1L and COMPASS there are:

- Xue et al, Nature 2019
- Hsu et al, Mol Cell 2019
- Park et al, Nat Comm 2019
- Anderson et al, Cell Rep 2019
- Valencia-Sanchez et al, Mol Cell 2019
- Jang et al, Genes Dev 2019
- Yao et al, Cell Res 2019

Please cite accordingly.

Response: We thank the reviewer to point out this issue. As the submission guideline for reference is up to 50, we are very sorry that some important works are not included in the original manuscript. We agree with the reviewer and have included the references as suggested in Page 11 in the revised manuscript.

REVIEWER COMMENTS

Reviewer #1 (Remarks to the Author):

The revised manuscript by Luo et al. has been considerably improved and contains important additional data. The authors convincingly replied to the suggestions and re-wrote the manuscript accordingly. I have no further comments/suggestions.

Reviewer #2 (Remarks to the Author):

The authors have addressed my main comments with further explanations, the inclusion of additional data and updated representations of gels and plots. In particular, they highlighted the interesting differences they observe between ubH2B as compared to their earlier work with ubH2A. They convincingly argue that structural changes within the nucleosome core are responsible for most differences they observe independent of FACT recruitment defects. However, they speculate that the location of ub on H2A might block FACT access. The authors could consider placing this discussion more prominently in their final manuscript.

Nevertheless, I leave these final minor revisions up to the authors and have no further comments preventing publication.

Reviewer #3 (Remarks to the Author):

In this revision, the authors added clarifying points on figures and additional experiments to address reviewer concerns. Their measurements on the impact of FACT on ubiquitinated H2B is quite convincing and has an interesting model proposed. However, their recent quantitative measurement of affinity between uH2B and unmodified nucleosomes raises some questions regarding their gel based data, and their later conclusions regarding recruitment of FACT in vivo. I believe the authors can address these with a few additional experiments and/or revision of their text to reflect a more modest interpretation of uH2B on FACT activity (less recruitment, more allosteric).

Major comments

The authors write:

"The resulting K_d for FACT binding is 16.4±5nM (mean ± SE) for H2B-nucleosome and 10.2±4nM (mean ± SE) for ubH2B-nucleosome (please find more calculation details in Methods). The results showed that the purified FACT has a much higher affinity for the ubH2B nucleosome, which agree well with the mono-nucleosomes pull-down assays."

This statement does not seem true. The K_d's are not only within 2-fold difference of each other, but also within the measured error bars. At first glance it appears they have near identical affinities. Can the authors please comment on why there is this large discrepancy between their measured biophysical data versus the data seen on gels/blots? What is the lower end of the sensitivity on their K_d measurement? Is the lack of significant difference due to bottoming out of the assay? Additionally, is it possible that there's a greater affinity difference between H2A-H2B vs H2A-uH2B dimers?

Comparison with known uH2B responsive proteins (DOT1L and COMPASS) the lack of affinity boosting by ubiquitin actually has literature precedent. In both the DOT1L and COMPASS cases, ubiquitin does

not play a recruitment/affinity role, but rather an allosteric role to promote enzyme activity.

Additionally, I echo the comments from Reviewer 2 regarding chaperone activity/uH2B recognition motifs as being highly important to decipher. In addition to dissecting the chaperoning activity from uH2B recognition, I think from the in vivo standpoint it is an open question whether FACT is really being recruited individually to uH2B nucleosomes or is a consequence of being associated with RNA Pol II during transcription. It would be useful for the authors to design mutations that disrupt RNA Pol II association to see if FACT is still being recruited to uH2B sites in the genome. With recent structures of Pol II and FACT on nucleosomes (PDB 7XTI) there appears to be a very defined surface between FACT subunits and the polymerase that can be targeted for mutagenesis.

While these experiments may be extensive, I believe they get to the key question of whether or not uH2B is actively serving a recruitment role or an allosteric one. Based on the in vitro biochemical data, I lean towards an allosteric model of uH2B action on FACT given the near identical K_d 's measured biophysically. In lieu of experiments, the authors can also revisit their interpretation of the data with regards to uH2B recruitment and update the text to reflect this.

H2B ubiquitination recruits FACT to maintain a stable altered nucleosome state for transcriptional activation.

Reviewer #1 (Remarks to the Author):

The revised manuscript by Luo et al. has been considerably improved and contains important additional data. The authors convincingly replied to the suggestions and re-wrote the manuscript accordingly. I have no further comments/suggestions.

Response: We thank the reviewer for the approval and positive comments on our revised manuscript.

Reviewer #2 (Remarks to the Author):

The authors have addressed my main comments with further explanations, the inclusion of additional data and updated representations of gels and plots. In particular, they highlighted the interesting differences they observe between ubH2B as compared to their earlier work with ubH2A. They convincingly argue that structural changes within the nucleosome core are responsible for most differences they observe independent of FACT recruitment defects. However, they speculate that the location of ub on H2A might block FACT access. The authors could consider placing this discussion more prominently in their final manuscript.

Nevertheless, I leave these final minor revisions up to the authors and have no further comments preventing publication.

Response: We thank the reviewer for the approval and positive comments on our revised manuscript. It's a very good suggestion to highlight the regulation of different structural changes within the nucleosome by ubH2A and ubH2B on FACT's function; we have revised the discussion section in the manuscript according to the reviewer's suggestion in revised Page 13.

Reviewer #3 (Remarks to the Author):

In this revision, the authors added clarifying points on figures and additional experiments to address reviewer concerns. Their measurements on the impact of FACT on ubiquitinated H2B is quite convincing and has an interesting model proposed. However, their recent quantitative measurement of affinity between uH2B and unmodified nucleosomes raises some questions regarding their gel based data, and their later conclusions regarding recruitment of FACT in vivo. I believe the authors can address these with a few additional experiments and/or revision of their text to reflect a more modest interpretation of uH2B on FACT activity (less recruitment, more allosteric).

Response: We thank the reviewer for all the positive comments and constructive suggestions. We have performed additional experiments to address the reviewer's concern and revised our manuscript as suggested, which are described as follows.

Major comments

The authors write: "The resulting K_d for FACT binding is 16.4 ± 5 nM (mean \pm SE) for H2B-nucleosome and 10.2 ± 4 nM (mean \pm SE) for ubH2B-nucleosome (please find more calculation details in Methods). The results showed that the purified FACT has a much higher affinity for the ubH2B nucleosome, which agree well with the mono-nucleosomes pull-down assays."

This statement does not seem true. The K_d 's are not only within 2-fold difference of each other, but also within the measured error bars. At first glance it appears they have near identical affinities. Can the authors please comment on why there is this large discrepancy between their measured biophysical data versus the data seen on gels/blots? What is the lower end of the sensitivity on their K_d measurement? Is the lack of significant difference due to bottoming out of the assay? Additionally, is it possible that there's a greater affinity difference between H2A-H2B vs H2A-uH2B dimers?

Comparison with known uH2B responsive proteins (DOT1L and COMPASS) the lack of affinity boosting by ubiquitin actually has literature precedent. In both the DOT1L and COMPASS cases, ubiquitin does not play a recruitment/affinity role, but rather an allosteric role to promote enzyme activity.

Response: The reviewer mainly concerned about the quantitative measurement of the binding affinity between ubH2B and unmodified nucleosomes on FACT.

Firstly, we carefully re-examined our quantitative measurement of K_d for FACT's binding. In order to determine the binding affinity of FACT for ubH2B and unmodified nucleosome more precisely, we performed the force-jump measurements by magnetic tweezers with fairly adequate stretching cycles for more different single-nucleosome samples. The different mechanical responses facilitate us to check the binding state of FACT on the substrates directly at the single molecule level as shown in the Figure a-b below. We showed the distribution of the K_d for FACT's binding to ubH2B or unmodified nucleosome with the significant difference analysis in the Figure c below. The K_d of FACT binding with the H2B-nucleosome is 16.3 ± 3.9 nM (mean \pm SE, 1000 stretching cycles for 30 samples), and the K_d of FACT binding with the ubH2B-nucleosome is 8.0 ± 4.2 nM (mean \pm SE, 980 stretching cycles for 30 samples), with significant difference $p < 0.01$. These results indicate that ubH2B do enhance the binding of FACT on nucleosome, with the K_d about 2-fold higher than that for the unmodified nucleosomes.

Secondly, we agree with the reviewer's suggestion that the allosteric effect of ubH2B on nucleosome may play an important role here. As we revealed in this paper (Figure 1), ubH2B impairs the stability of nucleosome. And we also have known that FACT prefers to bind the partially unwrapped nucleosome (*Mol Cell*. 2021, 81(17): 3542-3559), the sub-nucleosome (*Nature* 2020, 577(7790): 426-431), or the nucleosome with double-strand break site (*Genes Dev*. 2016, 30(6): 673-86), rather than the intact nucleosome. The allosteric effect of ubH2B on

nucleosome may lead to impair the nucleosome stability and also affect the function of FACT on nucleosome. But in our quantitative measurement of K_d by using the magnetic tweezers, for both ubH2B- and unmodified nucleosome, we measured the data by exerting forces to open the DNA wrap of nucleosome (with the detailed processes shown in methods), which may more or less conceal the allosteric effect of ubH2B on nucleosome. We believed it may be one of the reasons for the discrepancy between the measured biophysical data vs the data seen on gels. We have added the related comments in the revised manuscript Page 6 and 13.

Thirdly, we also quantified the effect of ubH2B on the property of FACT-mediated nucleosome assembly. The deposition ability of FACT for H2A-H2B and H2A-ubH2B dimer onto tetrasome was further quantitatively analyzed by magnetic tweezers. After the tethered tetrasome was incubated in FACT and H2A-H2B or H2A-ubH2B mixture for 30 minutes, we performed the force-extension measurements and analyzed the samples to quantify the numbers of formed nucleosomes and the remained tetrasomes. We found that in the presence of H2A-H2B dimer, $54.5 \pm 5.3\%$ (mean \pm SE, 150 samples were measured in three independent groups) of the tetrasomes were successfully assembled by FACT to form nucleosome structure.

In the presence of H2A-ubH2B dimer, $79.1 \pm 3.5\%$ (mean \pm SE, 150 samples were measured in three independent groups) of the tetrasomes were assembled to nucleosome. The results confirmed that ubH2B facilitates FACT to deposit H2A-H2B dimers onto H3-H4 tetramer to form ubH2B nucleosome.

In summary, our results showed that ubH2B not only promotes the recruitment of FACT on nucleosome state through direct binding or allosteric effect of ubH2B, but also enhance FACT to deposit H2A-H2B onto tetrasome to form ubH2B-nucleosome, which may both contribute to the recruitment and the coordination of FACT with ubH2B *in vivo*, as we proposed in the model in Figure 5. We have added the figures in revised Figure S3i and Figure S5d, and add the related comments in the revised manuscript Page 6, 8 and 13.

Additionally, I echo the comments from Reveiwer 2 regarding chaperone activity/H2B recognition motifs as being highly important to decipher. In addition to dissecting the chaperoning activity from uH2B recognition, I think from the *in vivo* standpoint it is an open question whether FACT is really being recruited individually to uH2B nucleosomes or is a consequence of being associated with RNA Pol II during transcription. It would be useful for the authors to design mutations that disrupt RNA Pol II association to see if FACT is still being recruited to uH2B sites in the genome. With recent structures of Pol II and FACT on nucleosomes (PDB 7XTI) there appears to be a very defined surface between FACT subunits and the polymerase that can be targeted for mutagenesis.

While these experiments may be extensive, I believe they get to the key question of whether or not uH2B is actively serving a recruitment role or an allosteric one. Based on the *in vitro* biochemical data, I lean towards an allosteric model of uH2B action on FACT given the near identical K_d 's measured biophysically. In lieu of experiments, the authors can also revisit their interpretation of the data with regards to uH2B recruitment and update the text to reflect this.

Response: We thank the reviewer to point out this issue. It's a very interesting question. We totally agree with the reviewer; it's of great interest to examine how ubH2B-nucleosome coordinates with RNA Pol II and FACT on transcription process. Actually, we are now trying to resolve the high-resolution cryo-EM structure of ubH2B-nucleosome bound with FACT and/or Pol II, and also trying to set up an *in vitro* transcription assay coupled with the *vivo* study as the review suggested to clear this question. BUT as the reviewer pointed out, these experiments are extensive and we don't think we can finish these kinds of experiments within

a short time frame. Definitely, it's a very important question, and we are going to address these important issues in our future works.

REVIEWERS' COMMENTS

Reviewer #3 (Remarks to the Author):

The authors have very earnestly addressed my concerns through additional experiments and discussion in the manuscript. While I believe the 2 fold effect in affinity of uH2B nucleosomes is significant statistically, it seems difficult to imagine it playing a dramatic role in recruitment biologically given the very tiny boost in binding. However, the authors have made significant changes in the text to reflect a more modest interpretation of the recruitment aspect of the model. I have no further suggestions or comments.

REVIEWERS' COMMENTS

Reviewer #3 (Remarks to the Author):

The authors have very earnestly addressed my concerns through additional experiments and discussion in the manuscript. While I believe the 2 fold effect in affinity of uH2B nucleosomes is significant statistically, it seems difficult to imagine it playing a dramatic role in recruitment biologically given the very tiny boost in binding. However, the authors have made significant changes in the text to reflect a more modest interpretation of the recruitment aspect of the model. I have no further suggestions or comments.

Response: We thank the reviewer for the approval and positive comments on our additional experiments and discussion in the revised manuscript.